# DIFF-PROMPT: DIFFUSION-DRIVEN PROMPT GENERATOR WITH MASK SUPERVISION

**Weicai Yan, Wang Lin, Zirun Guo, Ye Wang, Fangming Feng, Xiaoda Yang, Zehan Wang, Tao Jin** [*]
Zhejiang University
{yanweicai,linwanglw,gzr,yew}@zju.edu.cn
{fangmingfeng,xiaodayang,wangzehan01,jint_zju}@zju.edu.cn

## ABSTRACT

Prompt learning has demonstrated promising results in fine-tuning pre-trained multimodal models. However, the performance improvement is limited when applied to more complex and fine-grained tasks. The reason is that most existing methods directly optimize the parameters involved in the prompt generation process through loss backpropagation, which constrains the richness and specificity of the prompt representations. In this paper, we propose **Diff**usion-Driven **Prompt** Generator (Diff-Prompt), aiming to use the diffusion model to generate rich and fine-grained prompt information for complex downstream tasks. Specifically, our approach consists of three stages. In the first stage, we train a Mask-VAE to compress the masks into latent space. In the second stage, we leverage an improved Diffusion Transformer (DiT) to train a prompt generator in the latent space, using the masks for supervision. In the third stage, we align the denoising process of the prompt generator with the pre-trained model in the semantic space, and use the generated prompts to fine-tune the model. We conduct experiments on a complex pixel-level downstream task, referring expression comprehension, and compare our method with various parameter-efficient fine-tuning approaches. Diff-Prompt achieves a maximum improvement of 8.87 in R@1 and 14.05 in R@5 compared to the foundation model and also outperforms other state-of-the-art methods across multiple metrics. The experimental results validate the effectiveness of our approach and highlight the potential of using generative models for prompt generation. Code is available at https://github.com/Kelvin-ywc/diff-prompt.

## 1 INTRODUCTION

Pre-trained multimodal models (Radford et al., 2021; Jia et al., 2021; Yuan et al., 2021; Pham et al., 2023; Li et al., 2022b; Zhang et al., 2022a; Li* et al., 2022) have received widespread attention due to their strong generalization capabilities. Taking the Contrastive Language-Image Pretraining (CLIP) (Radford et al., 2021) as an example, it is pre-trained on web-scale data, which enables it to learn joint vision-language representations. Fine-tuning techniques enable these models to be effectively applied to downstream tasks. Early approaches utilized full fine-tuning; however, these methods demands considerable computational resources and compromise the generalization capabilities of the pre-trained model.

Prompt learning (Lester et al., 2021; Jia et al., 2022; Zhou et al., 2022b; Khattak et al., 2023a; Zhang et al., 2024a), as an efficient fine-tuning method, has garnered extensive research interest. It involves designing prompts either manually or automatically for fine-tuning pre-trained models. The advantages include significantly reducing training resources while preserving the original generalization capabilities. As shown in Fig. 1(a), most current prompt learning methods follow the first two paradigms. For the first paradigm (Jia et al., 2022; Zhou et al., 2022b; Wang et al., 2022a; MA et al., 2023; Fang et al., 2023), learnable prompts are added to the encoder input of the pre-trained model. These approaches have certain limitations: first, prompts for different modalities are learned independently, preventing the establishment of inter-modal connections. Second, only global prompts

---

[*]Corresponding author.

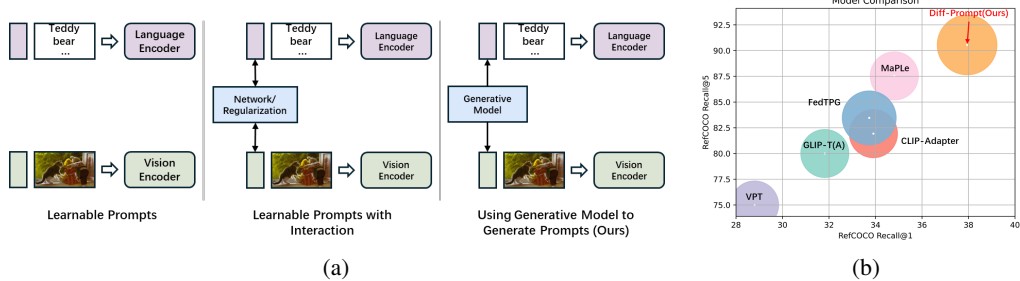

Figure 1: (a) Comparison between mainstream prompt learning methods (the first two paradigms) and our Diff-prompt paradigm. (b) Comparison of different efficient fine-tuning methods on the RefCOCO dataset, with the x-axis representing R@1, the y-axis representing R@5, and bubble size indicating the total model parameters. Diff-Prompt achieves higher performance at the cost of using partial parameters.

can be learned for all training data, which restricts the prompting capability. Subsequent works follow the second paradigm for improvements (Khattak et al., 2023a; Shi et al., 2024; Qiu et al., 2024; Roy & Etemad, 2024), using networks or regularization methods to establish connections between prompts of different modalities. However, we believe that the above methods update prompts or the prompt generation process in a goal-driven manner, which significantly limits the richness of the prompts. When applied to complex and fine-grained downstream tasks, their prompting capability is limited. As shown in Fig. 1(b), for a multimodal localization task that requires consideration of the complex relationships between modalities and multimodal understanding, VPT (Jia et al., 2022) adds prompts only on the visual modality side, and its performance is even inferior to that of the foundation model. Other prompt learning methods also show limited performance improvement.

To address the above issues, we consider how to generate rich prompts that can provide sufficient information to the pre-trained model, even for fine-grained downstream tasks. Inspired by the powerful feature extraction and generation capabilities of diffusion models, this paper proposes Diff-Prompt, which uses the diffusion model to generate rich prompt information. The training process of the diffusion model employs masks as supervision to inform the model which parts of the input image need to be emphasized for a given caption. Specifically, Diff-Prompt consists of three stages. In the first stage, we map the masks to latent space, extracting dense information while reducing computational load in the later stages. In the second stage, we train a prompt generator with mask supervision in latent space using an improved DiT model, conditioned on the image and caption to generate emphasized parts of the image. In the third stage, we align the prompt generator's output with the pre-trained model semantically to better integrate the generated prompts into the pre-trained model. Finally, we concatenate the generated prompts with a few learnable global prompts to supplement universial knowledge. We conduct experiments on a fine-grained multimodal task, specifically the referring expression comprehension, and evaluate on multiple metrics. The results demonstrate that our method outperforms other existing efficient fine-tuning methods, validating its effectiveness.

Our main contributions are as follows: (1) We train a Mask-VAE and a diffusion-driven prompt generator to generate rich prompt information in the mask latent space. (2) We align the generated prompts with the pre-trained model in the semantic space to effectively guide the pre-trained model. (3) We conduct experiments on a complex fine-grained mutli-modal downstream task, and the experimental results demonstrate the effectiveness of our method.

## 2 RELATED WORK

### 2.1 VISION-LANGUAGE MODELS

Vision-language models trained on large-scale data exhibit strong feature extraction and generalization capabilities. These models include CLIP (Radford et al., 2021), ALIGN (Jia et al., 2021), Florence (Yuan et al., 2021), BASIC (Pham et al., 2023), and OpenCLIP (Ilharco et al., 2021). When addressing downstream tasks, they are considered ideal choices. Additionally, some works pro-

pose pre-trained models for specific tasks, allowing for easy transfer to particular data distribution. LSeg (Li et al., 2022), and CLIPSeg (Lüddecke & Ecker, 2022) are used for segmentation, BLIP (Li et al., 2022a) is used for visual question answering, while GLIP (Li et al., 2022b), PPMN (Ding et al., 2022) and Grounding DINO (Liu et al., 2023) are used for localization tasks.

## 2.2 PROMPT TUNING

Prompt learning is initially applied in the field of natural language processing (NLP)(Petroni et al., 2019; Brown et al., 2020; Wallace et al., 2019; Shin et al., 2020; Li & Liang, 2021; Lester et al., 2021), where it achieves excellent performance. The core idea is to design manually crafted or automatically learned prompts to fine-tune pre-trained models. This approach allows pre-trained models to adapt to downstream tasks while avoiding the excessive resource consumption that comes with fully fine-tuning the models. Subsequently, prompt learning has been widely applied to the fields of computer vision (CV) (Jia et al., 2022; Bahng et al., 2022) and multimodal learning (Zhou et al., 2022b;a; Zang et al., 2022; Khattak et al., 2023a; Cao et al., 2023; Guo et al., 2024a; Fu et al., 2024; Qiu et al., 2024; Guo et al., 2024b; Yang et al., 2024b; Yan et al., 2024; Jin et al., 2024; Li et al., 2024a; Shi et al., 2024; Roy & Etemad, 2024). VPT (Jia et al., 2022) concatenates learnable prompts to the input of the vision encoder layer, while CoOp (Zhou et al., 2022b) concatenates learnable prompts to the input of the language encoder. These works incorporate prompts only within a single modality. To enable communication between modalities, MaPLe (Khattak et al., 2023a) and UPT (Zang et al., 2022) introduce prompts for different modalities and establish connections between the prompts of these modalities. Recent work attempts to generate input-specific prompts. QNet (Shi et al., 2024) generates prompts using Quaternion Networks. Additionally, more work explore broader application scenarios for prompt learning. For example, L2P (Wang et al., 2022c) and S-prompts (Wang et al., 2022a) investigate the performance of prompt learning in continual learning. TPT (Shu et al., 2022) and PromptAlign (Hassan et al., 2023) applies prompt learning in the context of test-time adaptation.

## 2.3 DIFFUSION MODELS

Diffusion Models are a type of generative model that generates new data by simulating a gradual reverse process of data distribution. DDPM (Ho et al., 2020) introduced a method for generating data through the stepwise addition and removal of noise. IDDPM (Nichol & Dhariwal, 2021) improved upon DDPM by employing more efficient training strategies and finer denoising steps. To enhance the generation efficiency of diffusion models, DDIM (Song et al., 2020) is a non-Markov diffusion model that allows skipping certain steps during inference, while LDM (Rombach et al., 2022) performs diffusion by mapping data into latent space. The subsequent work, DiT (Peebles & Xie, 2023a), combines diffusion models with transformer architecture, leveraging the strong representational capabilities of transformers to improve generation quality. In terms of specific tasks, ControlNet (Zhang et al., 2023) is a type of controllable generative model that introduces additional conditional information to regulate the generation process. Works (Ruiz et al., 2023; Gal et al., 2022; Lin et al., 2024a;b) investigate the customization of diffusion models.

## 3 PRELIMINARY

### 3.1 PROMBLEM FORMULATION AND FOUNDATION MODEL

Given an image $v$ and a caption $q$, the objective of the task is to predict the location $o$ of the described object within the image. GLIP (Li et al., 2022b) is used as the foundation model, which primarily consists of a vision encoder $\text{Enc}_v(\cdot)$, a language encoder $\text{Enc}_l(\cdot)$, and a downstream head $\text{Head}(\cdot)$. The image $v$ is first divided into multiple patches, which are then embedded into $E_0^v$. The caption $q$ is tokenized and embedded into $E_0^l$. These embeddings are subsequently fed into the modality encoder to generate the corresponding modality features. These features are then passed to the downstream head to predict bounding boxes $\tilde{o}$ for referring objects. For GLIP, the downstream head is a region proposal network (RPN). RPN uses a sliding window to generate multiple candidate regions and then adjusts the positions and sizes of these anchor boxes to generate high-quality candidate regions. The GLIP training loss $\mathcal{L}_{base}$ is the sum of the classification loss $\mathcal{L}_{cls}$ and the localization loss $\mathcal{L}_{loc}$.

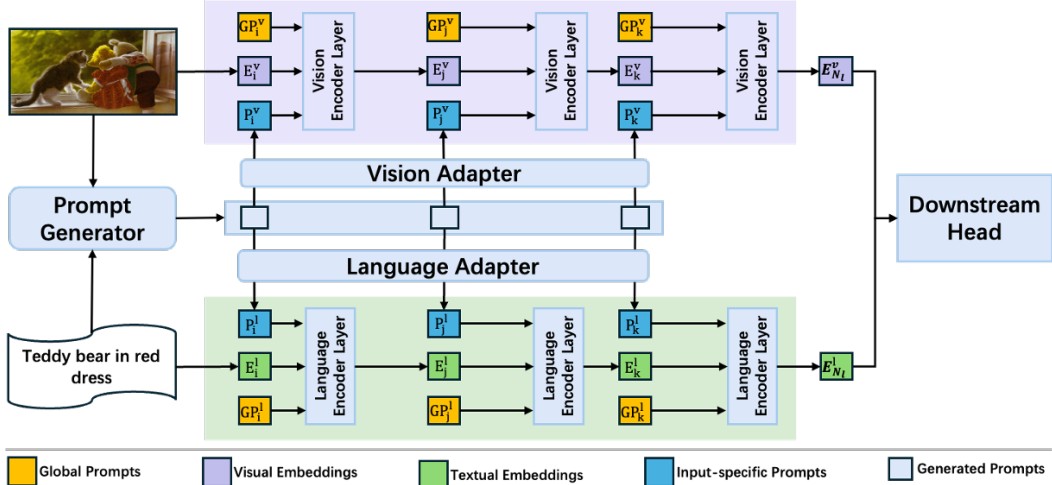

Figure 2: The framework of Diffusion-Driven Prompt Generator (Diff-Prompt). We fully utilize a diffusion model as the prompt generator, which generates prompts conditioned on a given image and caption. The generated prompts are then mapped into input-specific prompts through modality-specific adapters. These input-specific prompts are concatenated with global prompts of equal length to form the final prompts, which are used to fine-tune the pre-trained model.

## 3.2 DEEP PROMPTING

For an encoder $\text{Enc}(\cdot)$ composed of $N_l$ stacked attention layers $\boldsymbol{L} = \{L_i\}_{i=0}^{N_l - 1}$, the deep prompting technique introduces prompts into the first $D$ attention layers. For the $i$th attention layer, the prompt $\boldsymbol{P}_i \in \mathbb{R}^{N_p \times d_p}$ is concatenated with the embedding $\boldsymbol{E}_i$ as the input, where $N_p$ denotes the prompt length and $d_p$ is the dimension size.

## 3.3 DIFFUSION MODELS

For computational efficiency, we typically use a VAE encoder $\mathcal{E}$ to compress the target, then train a diffusion model in the latent space, and finally use a VAE decoder $\mathcal{D}$ to reconstruct the generated target. Our training objective is to obtain a diffusion model $\epsilon_\theta$ that can predict noise based on a given condition $C$. During inference, Gaussian noise is randomly sampled, and multi-step denoising is applied to obtain the generated result. Additionally, the skip-step strategy from DDIM is used to accelerate the generation process.

## 4 DIFF-PROMPT: DIFFUSION-DRIVEN PROMPT GENERATOR

We propose the Diff-Prompt, which aims to efficiently fine-tune pre-trained foundation models using prompts generated by a diffusion model. Diff-Prompt consists of three stages. In the first stage, we train a Mask-VAE to compress the mask into a low-dimensional space. In the second stage, we use an enhanced DiT (Peebles & Xie, 2023b) as the prompt generator, generating prompts (denoted as generated prompts) given an image and caption. In the third stage, we freeze the backbone network, Mask-VAE, and the prompt generator trained in the first two stage, and design modality-specific adapters to align the latent features generated by prompt generator with the foundation model, mapping the generated prompts to the representations of the foundation model. We then introduce a small number of learnable global prompts to complement universal knowledge, thus generating more expressive features.

### 4.1 MASK-VAE TRAINING

In the first stage, we aim to use the DiT model with mask supervision to generate visual prompts that locate the approximate position of the object referenced by the caption in the image, thereby

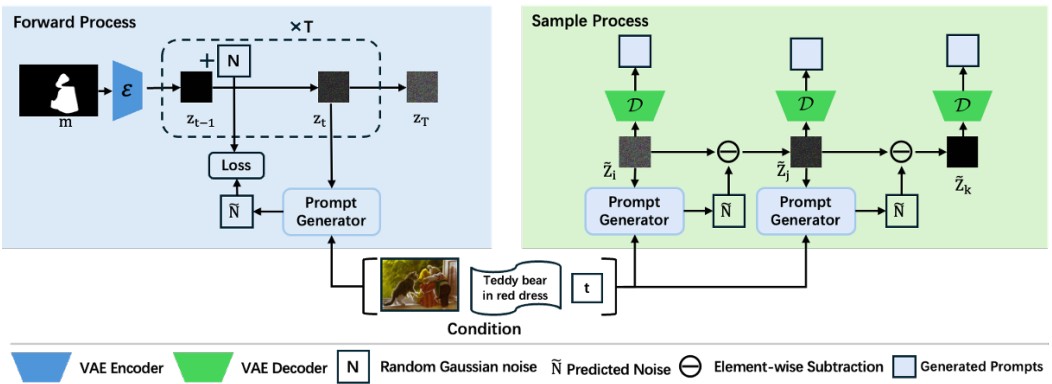

Figure 3: Forward and Sample Process of the Prompt Generator.

aiding the foundation model in reasoning. To reduce computational complexity, we follow the LDM approach by first training a Mask-VAE to compress the masks into the latent space. Given mask $m \in \mathbb{R}^{1 \times H \times W}$, where $H$ denotes the height and $W$ denotes the width, we train an encoder $\mathcal{E}$ and a decoder $\mathcal{D}$. The encoder $E$ encodes $m$ into a mean and variance vector in the latent space, and then the latent feature $z$ is sampled from the Gaussian distribution:

$$\mu, \sigma = \mathcal{E}(m), \quad z \sim \mathcal{N}(\mu, \sigma^2). \tag{1}$$

The decoder $\mathcal{D}$ then reconstructs the latent feature back to the original mask $\tilde{m}$:

$$\tilde{m} = \mathcal{D}(z). \tag{2}$$

To train the Mask-VAE, we use the following loss function:

$$\mathcal{L}_{vae} = \|m - \tilde{m}\|_2^2 + \lambda D_{KL}(\mathcal{N}(\mu, \sigma^2)\|\mathcal{N}(0, \mathbf{I})), \tag{3}$$

where $\lambda$ is the scale parameter.

## 4.2 VISUAL PROMPT GENERATION USING DIFFUSION MODEL

In the second stage, given an input image $v$ and caption $q$, we trained a prompt generator $\epsilon_\theta$ using the mask $m$ as guidance. For the diffusion process, the mask $m$ is first compressed into a latent feature $z_0$ by the encoder $\mathcal{E}$. We continuously add noise to $z$, repeating $T_{forward}$ times until it completely becomes Gaussian noise:

$$z_t = \sqrt{\bar{\alpha}_t z_0} + \sqrt{1 - \bar{\alpha}_t}\epsilon_t, \quad \epsilon_t \sim \mathcal{N}(0, \mathbf{I}). \tag{4}$$

The following loss is used to train the prompt generator $\epsilon_\theta$ with condition $C$:

$$\mathcal{L}_\theta = \|\epsilon_\theta(z_t, C) - \epsilon_t\|_2^2, \quad C = [\text{Emb}_v(v), \text{Emb}_q(q), \text{Emb}_t(t)], \tag{5}$$

where $\epsilon_\theta(z_t, C)$ is the predicted noise, $\text{Emb}_v(\cdot)$ is the image embedding layer, $\text{Emb}_q(\cdot)$ is the language embedding layer, $\text{Emb}_t(\cdot)$ is the timestep embedding layer, and $t$ is the timestep.

For the generation process, to prevent the prompt generation from taking too much time, we choose to use DDIM for accelerated sampling. The reverse process goes through $T_{sample}$ timesteps:

$$\tilde{z}_{t-1} = \tilde{z}_t - \epsilon_\theta(z_t, C). \tag{6}$$

As the diffusion steps of the prompt generator increase, the fusion of text and image information deepens. The latent features at the intermediate steps can effectively capture the fusion between different modalities, which is consistent with the encoding process. By aligning the diffusion process of the prompt generator with the encoding process of the pre-trained model's encoder, we can effectively inject modality fusion information into the encoder. Throughout the diffusion process of the prompt generator, we retain $D$ latent features $\mathbf{z} = [\tilde{z}_{i_0}, \tilde{z}_{i_1}, \ldots, \tilde{z}_{i_{D-1}}]$, where $\{i_0, i_1, \ldots, i_{D-1}\} \subseteq \{0, 1, \ldots, T_{sample}\}$. These generated prompts represent the degree of modality information fusion, and are subsequently mapped to input-specific prompts and integrated into the pre-trained model's encoder.

### 4.3 Visual Prompt Tuning with Foundational Models

In the second stage, we retain the latent features from the denoising process of the prompt generator as prompts. Considering that the prompt generator and GLIP are in different semantic spaces, we first use the Mask-VAE decoder in the second stage to reconstruct the latent prompts, generating prompts of the same size as the image. These generated prompts form a saliency map, which informs the foundation model about which parts of the image require more attention. Furthermore, to integrate the generated prompts into the pre-trained model, we design an adapter for each modality, namely $\text{Adapter}_v$ and $\text{Adapter}_l$. The modality-specific adapter aligns the generated prompts with the space of the modality encoder. This design not only enables cross-modality prompting over time but also facilitates communication between different modalities:

$$\boldsymbol{P}_j^v = \text{Adapter}_v(\mathcal{D}(\tilde{z}_{i_j})), \quad \boldsymbol{P}_j^l = \text{Adapter}_l(\mathcal{D}(\tilde{z}_{i_j})), \quad j = 0, 1, \dots, D - 1. \tag{7}$$

The input-specific prompts are designed to tailor the prompts to the input data. At the same time, we added the same number of learnable global prompts, namely global visual prompts, $\{\boldsymbol{GP}_j^v\}_{j=0}^{D-1}$, and global textual prompts, $\{\boldsymbol{GP}_j^l\}_{j=0}^{D-1}$. The input-specific prompts, global prompts, and embeddings are concatenated and fed into the encoder, which ultimately produces the model's output:

$$[\_, \_, \boldsymbol{E}_{j+1}^m] = L_j^m([\boldsymbol{P}_j^m, \boldsymbol{GP}_j^m, \boldsymbol{E}_j^m]), \quad j = 0, 1, \dots, D - 1, \tag{8}$$

$$[\boldsymbol{E}_{j+1}^m] = L_j^m([\boldsymbol{E}_j^m]), \quad j = D, \dots, N_l - 1, \tag{9}$$

$$\tilde{o} = \text{Head}(\boldsymbol{E}_{N_l}^v, \boldsymbol{E}_{N_l}^t), \tag{10}$$

where $m$ represents v(ision) or l(anguage) modality.

Throughout the entire third stage, we only train the parameters in the adapter and global prompts. To train our model, we select the same loss function as the one used in the foundation model.

## 5 Experiment

In this section, we first introduce the experiment setup, followed by a quantitative analysis. Next, we conduct qualitative analysis, ablation studies, and in-depth analysis on the RefCOCO val dataset. In Appendix D, we explore the zero-shot capabilities of Diff-Prompt. In Appendix E, we provide more visualization results compared with other methods. In Appendix C, We conduct an ablation experiment on prompt selection, and in Appendix F, we carry out further in-depth analysis on Flickr30k. Finally, we discuss the limitations in Appendix H.

### 5.1 Experiment Setup

**Dataset.** We conducted experiments on two vision-language understanding datasets, RefCOCO (Kazemzadeh et al., 2014) and Flickr30k (Plummer et al., 2016). RefCOCO includes a training set, two test sets (testA and testB), and a validation set (val). TestA contains multiple people, while testB contains multiple non-human objects. The Flickr30k dataset includes the train, test, and val set.

**Evaluation Metrics.** We use Recall at K (R@K) and Upper Bound (UB) as the evaluation metric. R@K indicates the proportion of times the model correctly identifies the target within the top K retrieval results, reflecting the model's ability to find the correct match within a given ranking range. UB evaluates the proportion of target presence among all prediction results. Specifically, we chose R@1, R@5 and UB as the evaluation standards. R@1 measures the model's ability for precise retrieval, R@5 reflects the model's overall recall ability, and UB indicates the model's potential.

**Baseline.** we conduct a quantitative analysis, selecting GLIP-T(A) (Li et al., 2022b) as the foundation model. We compare our model with two efficient parameter tuning methods: adapter and prompt tuning. For the adapter method, we choose Tip-adapter (Zhang et al., 2022b), Meta-adapter (Song et al., 2023), CLIP-Adapter(Gao et al., 2024), MMA(Yang et al., 2024a) for comparison. For MMA, we introduce the adapter in the last Transformer layer of the encoder. For Tip-Adapter, Meta-adapter, and CLIP-Adapter, we introduce the adapter at the encoder's output. For prompt tuning, we select VPT(Jia et al., 2022), VFPT (Zeng et al., 2024), CoOp(Zhou et al., 2022b), S-Prompts(Wang et al., 2022b), MaPLe(Khattak et al., 2023a), and FedTPG(Qiu et al., 2024).

Table 1: Performance Evaluation on RefCOCO and Flickr30k datasets. **Bold**: best results, underline: second best results.

| Method | RefCOCO (testA) | | | RefCOCO (testB) | | | RefCOCO (val) | | | Flickr30k (test) | | | Flickr30k (val) | | |
|---|---|---|---|---|---|---|---|---|---|---|---|---|---|---|---|
| | R@1 | R@5 | UB | R@1 | R@5 | UB | R@1 | R@5 | UB | R@1 | R@5 | UB | R@1 | R@5 | UB |
| GLIP-T(A) | 30.21 | 80.66 | 93.44 | 32.93 | 77.66 | 88.30 | 31.82 | 80.00 | 91.42 | 45.57 | 63.72 | 70.55 | 44.87 | 63.40 | 70.17 |
| Tip-adapter | 34.68 | 86.24 | 97.25 | 32.83 | 78.39 | 94.02 | 34.56 | 83.06 | 95.47 | 50.16 | 74.89 | 84.53 | 48.22 | 73.54 | 85.19 |
| CLIP-Adapter | 34.08 | 85.81 | 98.18 | 32.76 | 76.72 | 93.39 | 33.91 | 81.93 | 96.24 | 49.30 | 73.12 | 84.78 | 47.15 | 71.72 | 83.24 |
| Meta-adapter | 35.02 | 87.96 | 98.64 | 33.29 | 78.67 | 95.14 | 36.54 | 85.09 | 96.34 | 51.32 | 75.36 | 85.16 | 49.18 | 74.87 | 88.14 |
| MMA | 36.68 | 89.03 | 99.13 | 34.67 | 79.06 | 95.88 | 35.28 | 86.46 | 97.18 | 52.60 | 77.04 | 85.77 | 51.43 | 76.28 | 89.46 |
| VPT | 29.64 | 80.40 | 89.09 | 27.65 | 71.29 | 81.04 | 28.80 | 75.00 | 84.56 | 44.27 | 70.19 | 83.68 | 43.83 | 70.19 | 83.69 |
| VFPT | 37.24 | 91.45 | 98.23 | 31.98 | 79.36 | 97.74 | 34.92 | 87.93 | 98.41 | 55.82 | 76.16 | 88.67 | 51.53 | 75.94 | 88.31 |
| CoOp | 36.89 | 93.18 | 99.61 | 32.29 | 82.89 | 97.21 | 35.31 | 88.62 | 98.01 | 51.29 | 74.85 | 88.32 | 50.95 | 74.79 | 87.96 |
| S-Prompts | 37.69 | 93.11 | 97.21 | 32.84 | 81.86 | 90.25 | 35.32 | 87.99 | 98.65 | 53.09 | 76.58 | 88.92 | 52.15 | 76.03 | 88.57 |
| MaPLe | 37.72 | 91.97 | 99.33 | 32.70 | 82.22 | 98.88 | 34.81 | 87.54 | 98.86 | 55.69 | 80.24 | **90.50** | 54.96 | 79.91 | 90.49 |
| FedTPG | 37.65 | 93.78 | 99.61 | 33.25 | 82.81 | 97.68 | 35.29 | 88.34 | 99.03 | 51.94 | 74.32 | 87.98 | 51.48 | 74.07 | 87.54 |
| FedTPG$_{d9}$ | 37.76 | 90.98 | 99.58 | 29.91 | 75.66 | 97.94 | 33.73 | 83.46 | 98.90 | 57.95 | 80.62 | 90.17 | 56.08 | 79.96 | 90.13 |
| Diff-Prompt | **39.08** | **94.71** | **99.63** | **36.09** | **85.67** | **99.00** | **37.94** | **90.55** | **99.37** | **59.53** | **81.85** | 90.46 | **57.39** | **81.20** | **90.54** |

**Experiment Detail.** For the Diff-Prompt, in the first stage, we train Mask-VAE on the RefCOCO dataset for 200 epochs, setting the batch size to 128, the learning rate to 0.05, and $\lambda$ to 0.0003. In the second stage, we train the prompt generator. During the training phase, we set $T_{forward} = 100$ and use squaredcos_cap_v2 as the noise scheduler. In the sampling phase, we use DDIM and set the number of sampling timesteps $T_{sample}$ to 25, with the batch size set to 128 and the number of epochs to 100. In the third stage, for the input of the $i$th attention layer, we select the latent features at step $25 - 2i$ as the generated prompts. The visual embedding size is set to 96, and the language embedding size is set to 768. The learning rate is set to 0.0001, and AdamW is used as the optimizer. In Appendix C, we discuss the rationale behind this choice. The specific architectures of Mask-VAE, the prompt generator and the adapters are detailed in the Appendix A. Additional experimental details are provided in Appendix B.1.

## 5.2 QUANTITATIVE ANALYSIS

The experimental results are shown in Tab. 1. What we can see from the results is that Diff-Prompt surpasses other adapter and prompt tuning methods across all metrics. Specifically, for the Ref-COCO dataset, Diff-Prompt shows performance improvements across all three subsets. Compared to the GLIP-T(A) model, Diff-Prompt achieves a maximum increase of 8.87% in R@1 and 14.05% in R@5 in testA dataset. Furthermore, we observed that the more interaction between prompts, the more significant the performance improvement. VPT and VFPT add prompt information only on the visual side, resulting in less performance improvement compared to methods that incorporate prompts in both modalities, such as MaPLe and FedTPG. Additionally, it is found that CLIPAdapter, CoOp, S-Prompts, MaPLe, and FedTPG all show improvements across metrics on the testA subset, but there is a slight decrease in performance on testB for some metrics. Based on the distribution of data in testA and testB, we can infer that the model overfits images in the person class during training, leading to a slight decline in performance for other classes. Notably, the CLIP-Adapter method outperforms VPT but falls short of CoOp. This is because CLIP-Adapter maps modality features but fails to provide additional auxiliary information to the pre-trained model, thus limiting the performance enhancement. The performance on Flickr30k is similar to that on RefCOCO. As the interaction between different modality prompts deepens, the richer the content of the prompts, the more significant the performance improvement.

Overall, prompt tuning methods generally achieve greater improvements in accuracy compared to adapter tuning. This is because adapter tuning typically requires adjusting the learned network to the entire data distribution, which may compromise the generalization ability of the original backbone network, making training more challenging. Consequently, the accuracy improvement is relatively limited. The performance boost of Diff-Prompt can be attributed to its ability to provide input-specific rich prompt information to the pre-trained model, leveraging the strong generative capabilities of the generative model based on image and caption content. In contrast, this is difficult to achieve with random initialization.

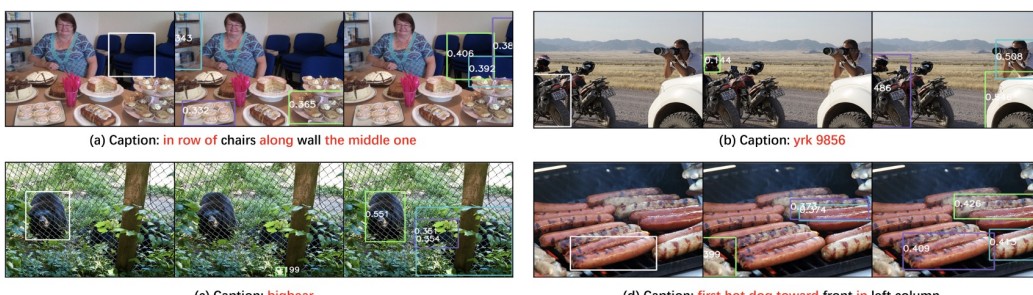

(a) Caption: in row of chairs along wall the middle one

(b) Caption: yrk 9856

(c) Caption: bigbear

(d) Caption: first hot dog toward front in left column

Figure 4: Qualitative Analysis for RefCOCO: Ground Truth (left), GLIP-T(A) (middle), Diff-Prompt (right). The results show the top three bounding boxes with the highest confidence, represented by green, blue, and purple from highest to lowest confidence, respectively. In the caption, the red content indicates positive tokens.

## 5.3 QUALITATIVE ANALYSIS

Visualization result is shown in Fig. 4, we can observe that: (1) Compared to the foundation model, Diff-Prompt is more sensitive to location; in figure (a), its attention is focused on the top right corner. (2) Diff-Prompt exhibits stronger language understanding; in figure (b), the caption refers to the motorcycle's license plate, but it understands that the license plate refers to the motorcycle itself. (3) Diff-Prompt has superior object recognition capabilities; as shown in figure (c), it can accurately identify occluded objects. (4) Diff-Prompt demonstrates stronger multimodal understanding, accurately identifying the referred object when multiple similar objects are present in the image.

## 5.4 ABLATION STUDY

We first conduct ablation studies to evaluate the generalization capability of our model. We follow prior work and conduct experiments on two benchmarks: the cross-dataset benchmark and the cross-domain benchmark. CoOp, MaPLe, PromptKD (Li et al., 2024b), PromptSRC (Khattak et al., 2023b), CoPrompt (Roy & Etemad, 2023), CPL (Zhang et al., 2024c), and CoCoLe (Zhang et al., 2024b) are chosen for comparison. The cross-dataset benchmark evaluates the model's generalization ability on shifted data. The cross-domain benchmark is illustrated in Appendix. D. Then, we ablate on the generalization ability across downstream tasks and backbones. Subsequently, we conduct ablation experiments on the effectiveness of prompts and prompt depth.

Table 2: Comparison with state-of-the-art methods on cross-dataset evaluation.

| Source | ImageNet | Caltech101 | OxfordPets | StanfordCars | Flowers102 | Food101 | Aircraft | SUN397 | DTD | EuroSAT | UCF101 | Average |
|---|---|---|---|---|---|---|---|---|---|---|---|---|
| CoOp | 71.51 | 93.70 | 89.14 | 64.51 | 68.71 | 85.30 | 18.47 | 64.15 | 41.92 | 46.39 | 66.55 | 63.88 |
| MaPLe | 70.72 | 93.53 | 90.49 | 65.57 | 72.23 | 86.20 | 24.74 | 67.01 | 46.49 | 48.06 | 68.69 | 66.30 |
| PromptKD | - | 93.61 | **91.59** | **73.93** | 75.33 | **88.84** | 26.24 | 68.57 | **55.08** | **63.74** | **76.39** | **71.33** |
| PromptSRC | 71.27 | 93.60 | 90.25 | 65.70 | 70.25 | 86.15 | 23.90 | 67.10 | 46.87 | 45.50 | 68.75 | 65.81 |
| CoPrompt | 70.80 | 94.50 | 90.73 | 65.67 | 72.30 | 86.43 | 24.00 | 67.57 | 47.07 | 51.90 | 69.73 | 67.00 |
| CPL | 73.53 | 95.52 | 91.64 | 66.17 | 73.35 | 87.68 | 27.36 | 68.24 | 48.96 | 51.25 | 70.52 | 68.07 |
| CoCoLe | **73.88** | **95.88** | 91.93 | 67.79 | 74.17 | 87.97 | 28.83 | 68.75 | 49.26 | 51.75 | 72.78 | 68.91 |
| Diff-Prompt | 72.06 | 94.63 | 91.08 | 66.85 | **75.57** | 87.26 | **29.07** | **69.03** | 48.87 | 52.83 | 73.32 | 68.85 |

**Cross-Dataset Generalization.** We consider the following 11 datasets to evaluate cross-domain performance: Aircraft (Maji et al., 2013), Caltech101 (Fei-Fei et al., 2004), Cars (Krause et al., 2013), DTD (Cimpoi et al., 2014), EuroSAT (Helber et al., 2019), Flower102 (Nilsback & Zisserman, 2008), Food101 (Bossard et al., 2014), Pets (Parkhi et al., 2012), SUN397 (Xiao et al., 2010), and UCF101 (Soomro, 2012). These datasets cover a wide range of categories, allowing us to assess the model's ability across diverse classes. The experimental results are shown in the Tab. 2. The conclusions for cross-dataset generalization and cross-domain generalization are similar. Although Diff-Prompt does not achieve state-of-the-art performance, it performs notably well on certain datasets, such as Flowers102, Aircraft, and SUN397.

| Method | mIoU | $IoU_{FG}$ | AP |
|---|---|---|---|
| CLIPSeg(PC) | 46.1 | 56.2 | 78.2 |
| CLIPSeg(PC, D=128) | 48.2 | 56.5 | 78.2 |
| CLIPSeg(PC)+Diff-Prompt | 47.8 | 56.4 | 78.2 |
| CLIPSeg(PC, D=128)+Diff-Prompt | 49.6 | 57.0 | 78.7 |

Table 3: Generalization Using CLIPSeg on the RES Task.

| Method | VQA | | NLVR[2] | |
|---|---|---|---|---|
| | test-dev | test-std | dev | test-P |
| BLIP | 78.24 | 78.17 | 82.48 | 83.08 |
| BLIP_CapFilt-L | 78.25 | 78.32 | 82.15 | 82.24 |
| BLIP+Diff-Prompt | 78.59 | 78.88 | 82.94 | 83.76 |
| BLIP_CapFilt-L+Diff-Prompt | 78.92 | 79.24 | 83.09 | 84.06 |

Table 4: Generalization Using BLIP on the GQA Task.

**Generalization Across Downstream Tasks and Backbones.** To validate the generality of Diff-Prompt, we conducted experiments using two different backbones on two new downstream tasks. Specifically, we employed the CLIPSeg (Lüddecke & Ecker, 2022) model on the PhaseCut dataset for the Referring Expression Segmentation task and the BLIP (Li et al., 2022a) model on the VQA-v2 and NLVR[2] datasets for the VQA task. Detailed experimental settings are provided in Appendix B.2, and the results are shown in Tab. 3 and 4. The results demonstrate that Diff-Prompt is equally effective with new backbones and downstream tasks. This effectiveness stems from our method's ability to guide the model to focus on relevant parts of the image based on captions, making it highly applicable to various visual understanding tasks.

**Effectiveness of Prompts.** We investigated the impact of different prompts by removing them one at a time: the visual prompt (w/o $P^v$), global visual prompt (w/o $GP^v$), textual prompt (w/o $P^l$), and global textual prompt (w/o $GP^l$). Results in Tab. 5 show that all prompts enhance accuracy. Removing task-specific prompts significantly reduces R@1 and R@5, highlighting their role in guiding the pretrained model. The textual prompt, derived from visual information, has the most substantial impact on accuracy when removed. This is because it integrates visual data into the language encoder, facilitating cross-modal interaction.

Table 5: Effectiveness of Prompts.

| Method | R@1 | R@5 | UB |
|---|---|---|---|
| Diff-Prompt | 37.94 | 90.55 | 99.37 |
| w/o $P^v$ | $36.94_{(-1.00)}$ | $89.84_{(-0.71)}$ | $99.15_{(-0.22)}$ |
| w/o $GP^v$ | $37.01_{(-0.93)}$ | $89.61_{(-0.94)}$ | $99.16_{(-0.21)}$ |
| w/o $P^l$ | $35.61_{(-2.33)}$ | $87.31_{(-3.24)}$ | $98.82_{(-0.55)}$ |
| w/o $GP^l$ | $36.95_{(-0.99)}$ | $89.81_{(-0.74)}$ | $99.11_{(-0.26)}$ |

**Prompt Depth.** This section investigates the impact of prompt depth. We selected prompt depths of 1, 3, 6, 9, and 12. Specifically, when the prompt depth is set to 1, prompts are added only at the encoder input, while for a depth of 12, prompts are added to the input of each transformer layer in the encoder. As shown in the Fig. 5, both R@1 and R@5 steadily increase as the prompt depth increases. When the prompt depth is shallow, the accuracy improvement is relatively slow, and there may even be a downward trend. However, as the depth increases, the improvement becomes more significant. This indicates that deeper prompts are more effective, likely because the deeper layers of the encoder encode richer information, facilitating easier information interaction. However, increasing the depth also leads to an increase in parameters and computational complexity, which is discussed in the complexity analysis.

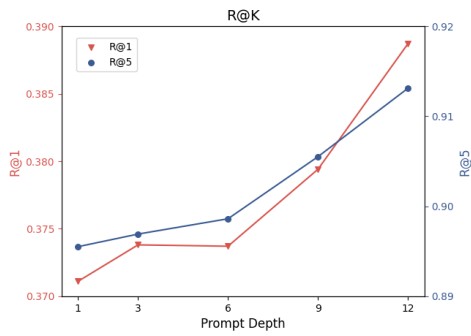

Figure 5: Metrics at Different Prompt Depths.

## 5.5 IN-DEPTH ANALYSIS

**Visual Prompt Visualization.** We visualize the prompts of the first 9 layers, which is shown in Fig. 6. Through progressive denoising, the visual and textual information is fully integrated. These prompts can provide information to the pre-trained model. Additionally, we observe that in the early stages of denoising, the prompts can already perceive the approximate location of the referent object. As the denoising process deepens, the prompts become increasingly informative. Although these prompts are not absolutely precise, they can provide fine-grained information and help filter out the approximate contours.

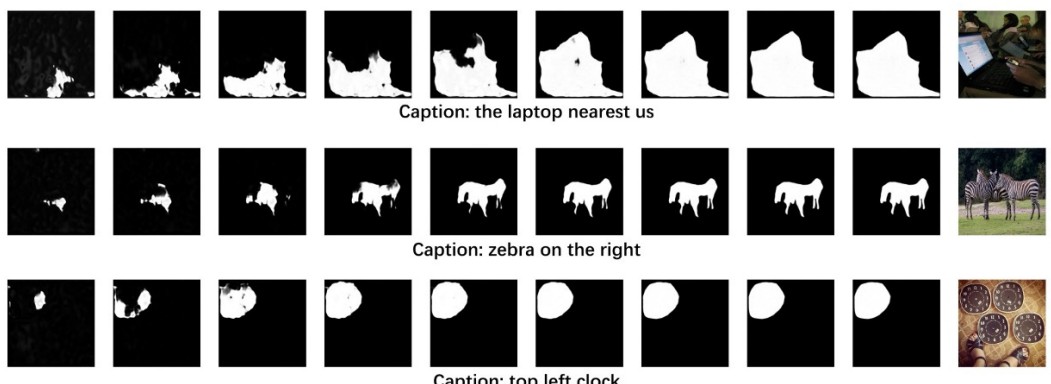

Figure 6: Visualization of Prompt Generation.

**Computation Complexity.** In this section, we explore the parameter introduction and computational complexity of different methods. We calculate the learnable parameters introduced by different models (# Tunable), the percentage of learnable parameters in the total model parameters (# Tunable %), computational complexity (Comp. Complex.) and inference time(Infer. Time). The results are shown in the Tab. 6. Regarding the introduction of parameters, VPT and CoOp only introduce a small

Table 6: Parameter and computational complexity analysis.

| Method | # Tunable | # Tunable % | Comp. Complex. (GFLOPs) | Infer. Time(s) |
|---|---|---|---|---|
| CLIPAdapter | 0.3M | 0.216 | 26.7 | 1.92 |
| VPT | 6.9K | 0.005 | 26.7 | 1.77 |
| CoOp | 55.3K | 0.037 | 27.1 | 1.85 |
| S-Prompts | 62.2K | 0.041 | 27.2 | 1.83 |
| MaPLe | 0.7M | 0.473 | 27.2 | 1.78 |
| FedTPG | 4.3M | 2.786 | 28.6 | 1.82 |
| $FedTPG_{d9}$ | 39.1M | 20.504 | 29.1 | 1.94 |
| Diff-Prompt | 5.5M | 4.834 | $28.2_{+7.7/Smp.}$ | 2.29 |

number of parameters, which are appended to the input of the attention layer. CLIP-Adapter and MaPLe require the introduction of additional network modules, leading to slightly more parameters compared to VPT and CoOp. For FedTPG and Diff-Prompt, these methods involve designing networks to generate prompts. To generate effective prompts, the network architecture is more complex than that of CLIP Adapter and MaPLe. For Diff-Prompt, Mask-VAE takes up 368kB, the Prompt Generator 309MB, while the foundation model GLIP-T(A) occupies 2.43GB. The additional parameters introduced by these prompt generators are still acceptable compared to the base model.

In terms of computational complexity, the complexity of other methods is roughly the same, while Diff-Prompt is much higher. This is because Diff-Prompt requires multi-step generation of prompts using a diffusion model. We optimized the model size and sampling speed as much as possible, resulting in a time complexity of 28.2 GFLOPs for the final model, plus 7.7 GFLOPs per sampling step. However, we found that in actual inference, the inference time of Diff-Prompt does not increase significantly, taking only 2.29 seconds. This is thanks to the transformer architecture of the diffusion model, which allows for high-speed parallel computation, and the diffusion model's size is an order of magnitude smaller than GLIP, thus not causing a significant increase in time.

# 6 CONCLUSION

This paper presents a new method for prompt generation based on a diffusion model. We find that, with appropriate supervision, the diffusion model can generate fine-grained prompts, achieving cross-modal information fusion and understanding. These generated prompts provide rich information that can help fine-tune pre-trained models for complex multimodal downstream tasks.

# 7 ACKNOWLEDGMENTS

This work was supported in part by Public Welfare Research Program of Ningbo under Grant No. 2024S062 and Yongjiang Talent Project of Ningbo under Grant No. 2024A-161-G.

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

## A    MODEL DESIGN

### A.1    MASK-VAE DESIGN

We use the AutoencoderKL class from the Python diffusers library to train our Mask-VAE, setting the in_channel parameter to 1. The Mask-VAE reads masks of size $1 \times 224 \times 224$ and compresses them into $4 \times 28 \times 28$. The final trained Mask-VAE occupies only 368kB in safetensors format.

### A.2    PROMPT GENERATOR ARCHITECTURE

The architecture of the prompt generator is generally the same as DiT, with only two differences. (1) the condition needs to incorporate both image and text information. We begin by using the same encoders as that in GLIP-T(A) to encode image and text, resulting in visual embedding, and textual embedding. The visual embedding and textual embedding are each added to timestep embedding, then concatenated to form the condition, which is fed into the DiT block. (2) For parameter selection, we aim to minimize the number of parameters while maintaining model performance. The number of DiT blocks is set to 12, the hidden size is set to 512, the patch size is 2, and the number of attention heads is set to 8. The final trained prompt generator occupies only 309MB when using float32 precision.

### A.3    ADAPTER DESIGN

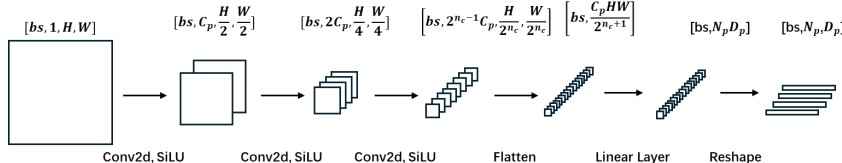

Figure 7: Adapter Architecture.

As shown in Fig. 7, the vision adapter and language adapter share the same network architecture. The latent features are first decoded through the mask-VAE decoder and then is mapped into input-specific prompts through the corresponding modality encoders. To avoid the significant resource waste that could result from direct mapping, we first reduce the dimension of the latent prompts using a few convolutional layers, and then perform the mapping in the low-dimensional space.

## B    EXPERIMENT DETAIL

### B.1    EXPERIMENTAL DETAILS OF COMPARATIVE EXPERIMENTS

VPT and VFPT introduces prompts into the visual encoder, while CoOp introduces prompts into the language encoder. S-Prompts, MaPLe, and FedTPG introduce prompts into both modalities simultaneously. It should be noted that S-Prompts are used in continual learning scenarios, while FedTPG is used in multiple remote clients scenarios. In this work, we only use their network architectures. The prompt length is set to 8 for all methods, while Diff-Prompt sets 4 input-specific prompts and 4 global prompts for each modality. Except for the FedTPG method, other prompt learning methods add prompts to the first 9 attention layers. Since FedTPG is originally designed to only add prompts at the encoder input, we introduce $\text{FedTPG}_{d9}$, which adds prompts to the first 9 layers of the encoder.

### B.2    EXPERIMENTAL DETAILS OF GENERALIZATION ABLATION EXPERIMENTS

We explore the generalization ability of Diff-Prompt across different backbones and tasks. Specifically, we select CLIPSeg for the Referring Expression Segmentation task and the BLIP model for the Visual Question Answering task. For the CLIPSeg model, its encoder is CLIP, with the original CLIP weights kept unchanged. Therefore, we adopt the same settings as the comparative experiments, introducing prompts at both the visual and textual encoder ends. For each modality, we use 4

Table 7: Results of different prompt strategies on the RefCOCO dataset.

| Strategy | testA | | | testB | | | val | | |
|---|---|---|---|---|---|---|---|---|---|
| | R@1 | R@5 | UB | R@1 | R@5 | UB | R@1 | R@5 | UB |
| sequential | 35.09 | 86.18 | 92.98 | 29.28 | 75.70 | 87.16 | 32.27 | 80.61 | 90.36 |
| reverse | 39.08 | 94.71 | 99.63 | 36.09 | 85.67 | 99.00 | 37.94 | 90.55 | 99.37 |

input-specific prompts and 4 global prompts. During training, only the adapter is trained, ensuring that the remaining network parameters remain unchanged.

For the BLIP model, which follows an encoder-decoder architecture, we introduce prompts only in the self-attention module of the encoder. During the training phase, we train only one adapter while keeping all other parameters unchanged.

## C PROMPT SELECTION

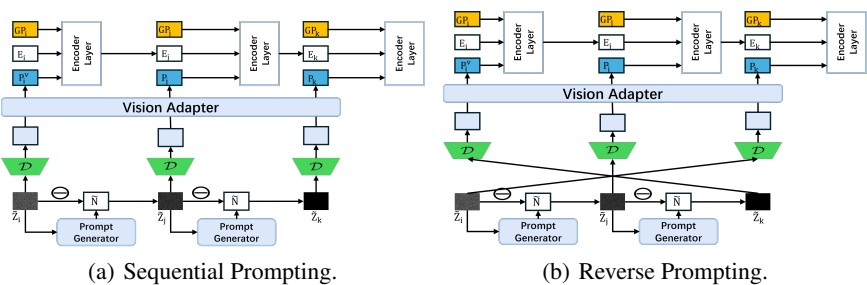

(a) Sequential Prompting.        (b) Reverse Prompting.

Figure 8: Different strategies for introducing prompts.

The prompt generator uses the image and caption as conditions, and the final result is obtained by denoising the random Gaussian noise 25 times. As the denoising process progresses, the image and text information gradually merge; that is, the denoising process can be seen as an interaction process between the image and text information. Therefore, as $t$ increases, the interaction between image and text increases. We choose $T_{sample} = 25$ to ensure the quality of the final result while minimizing the number of sampling steps and aligning with the pre-trained model. Here, we can align the sampling process with the encoding process either sequentially or in reverse order. In the sequential process Fig. 8(a), a small amount of interaction information is provided in the early stages of encoding, while in the reverse process Fig. 8(b), richer interaction information is provided in the shallow layers of the encoder. We conducted ablation experiments, and the results are shown in the Tab. 7. From the figure, we can see that introducing prompts in reverse order results in better performance. This improvement is likely due to incorporating more interactive information in the shallow layers of the encoder, which may better assist the encoding process.

## D ZERO-SHOT EVALUATION

This section explores the zero-shot capabilities of Diff-Prompt. We compare Diff-Prompt with GLIP-T(A) and GLIP-L. Compared to GLIP-T(A), GLIP-L has a larger model size, more training data, and stronger generalization abilities. We selected 11 representative datasets from ODinW (Li et al., 2022b) for testing: AmericanSignLanguageLetters (Letters), BCCD, brackishUnderwater (Underwater), CottontailRabbits (Rabbits), NorthAmericaMushrooms (Mushrooms), Packages, pistols, Raccoon, ShellfishOpenImages (Shellfish), thermalDogsAndPeople (DogsPeople), and VehiclesOpenImages (Vehicles). We use Average Precision (AP) @[IoU=0.5:0.95] as the metric.

**Zero-Shot Evaluation for Object Detection.** As shown in Tab. 8 and 9, the zero-shot capabilities of Diff-Prompt and foundation models are compared. From the figures, we can see that Diff-Prompt

Table 8: Zero-shot Evaluation of Diff-Prompt and Foundation models. (Part 1)

| | Letters | BCCD | Underwater | Rabbits | Mushrooms | Package |
|---|---|---|---|---|---|---|
| GLIP-T(A) | 0.07 | 5.83 | 0.53 | 65.27 | 29.72 | 32.43 |
| GLIP-L | 1.56 | 2.78 | 0.98 | 78.25 | 57.41 | 51.13 |
| Diff-Prompt | 2.25 | 8.42 | 1.21 | 71.27 | 54.18 | 41.74 |

Table 9: Zero-shot Evaluation of Diff-Prompt and Foundation models. (Part 2)

| | pistols | Raccoon | Shellfish | DogsPeople | Vehicles | Average |
|---|---|---|---|---|---|---|
| GLIP-T(A) | 32.28 | 15.43 | 15.34 | 40.82 | 45.35 | 25.73 |
| GLIP-L | 71.5 | 47.96 | 46.93 | 64.82 | 55.75 | 43.55 |
| Diff-Prompt | 12.82 | 45.63 | 11.3 | 34.27 | 20.22 | 27.57 |

retains the generalization ability of GLIP-T(A), and even outperforms GLIP-T(A) on a portion of the datasets. This is because Diff-Prompt provides additional auxiliary information for the original GLIP-T(A) without making any changes to the model itself. However, compared to GLIP-L, which has a larger parameter size and more training data, Diff-Prompt and GLIP-L still have a significant gap. This indicates that prompt learning's improvement to performance is still limited. Notably, for the Letters and Underwater datasets, both GLIP-T(A) and GLIP-L perform particularly poorly. In contrast, Diff-Prompt shows a subtle improvement, suggesting that when the pretrained model fails to extract feature information effectively, prompt information can play a significant role in enhancing performance.

Table 10: Comparison with state-of-the-art methods on cross-domain evaluation.

| Source | ImageNet | -V2 | -S | -A | -R | Avg. |
|---|---|---|---|---|---|---|
| CLIP | 66.73 | 60.83 | 46.15 | 47.77 | 73.96 | 57.18 |
| CoOp | 71.51 | 64.20 | 47.99 | 49.71 | 75.21 | 59.28 |
| MaPLe | 70.72 | 64.07 | 49.15 | 50.90 | 76.98 | 60.27 |
| PromptKD | - | - | - | - | - | - |
| PromptSRC | 71.27 | 64.35 | 49.55 | 50.90 | 77.80 | 60.65 |
| CoPrompt | 70.80 | 64.25 | 49.43 | 50.50 | 77.51 | 60.42 |
| CPL | 73.53 | 65.18 | 49.92 | 50.73 | 77.38 | 60.80 |
| CoCoLe | **73.88** | **65.86** | 50.89 | **51.75** | **78.89** | **61.85** |
| Diff-Prompt | 72.06 | 64.29 | **51.06** | 50.97 | 77.18 | 60.88 |

**Cross-Domain Generalization.** Ablation study for cross-domain generalization assesses the model's capability across different categories. We select ImageNet (Deng et al., 2009) and its four variations: ImageNet-A (Hendrycks et al., 2021b), ImageNet-V2 (Recht et al., 2019), ImageNet-R (Hendrycks et al., 2021a) and ImageNet-S (Wang et al., 2019), as the evaluation datasets. Specifically, we can consider that the CLIP model is well-fitted to the ImageNet dataset, while the data from the other four datasets are treated as out-of-distribution. As shown in the Tab. 10, Diff-Prompt achieves competitive accuracy with CPL and outperforms most prompt-learning methods, though it still lags behind CoCoLe. Notably, it achieves the best performance on ImageNet-S, highlighting its robustness against overfitting due to its controlled, generated prompts, unlike directly learned prompts, which are more prone to overfitting.

# E  QUALITATIVE ANALYSIS

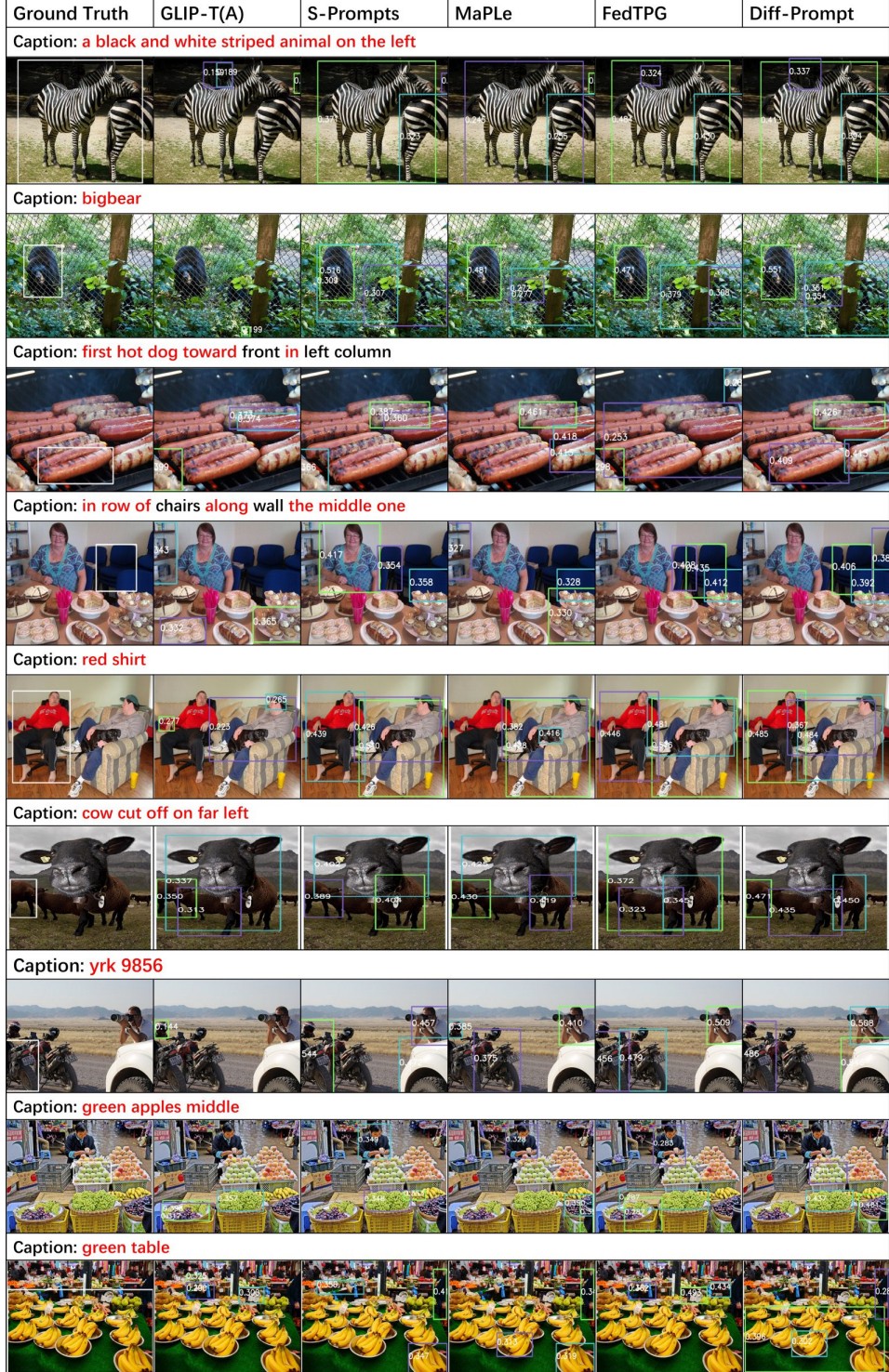

Figure 9: Qualitative Analysis on RefCOCO.

This section provides additional visualization results. We visualize Ground Truth, GLIP-T(A), S-Prompts, MaPLe, FedTPG, and Diff-prompt. From the Fig. 9, it can be seen that Diff-Prompt outperforms the other methods on most of the situation.

# F IN-DEPTH ANALYSIS OF THE FLICKR30K

| Method | Animals | | | | Bodyparts | | | | Clothing | | | | Instruments | | | |
|---|---|---|---|---|---|---|---|---|---|---|---|---|---|---|---|---|
| | R@1 | R@5 | R@10 | UB | R@1 | R@5 | R@10 | UB | R@1 | R@5 | R@10 | UB | R@1 | R@5 | R@10 | UB |
| CLIP-Adapter | 78.20 | 91.78 | 93.69 | 94.07 | 6.28 | 16.82 | 21.07 | 35.49 | 26.60 | 50.32 | 60.04 | 71.31 | 43.87 | 65.81 | 78.06 | 82.58 |
| VPT | 66.54 | 88.53 | 91.20 | 93.12 | 11.46 | 25.69 | 32.72 | 46.21 | 24.84 | 46.38 | 58.37 | 74.48 | 35.48 | 57.42 | 60.00 | 69.68 |
| CoOp | 78.39 | **95.79** | 95.79 | 96.56 | 7.76 | 18.48 | 27.17 | 44.73 | 31.09 | 57.04 | 67.88 | 79.79 | 48.39 | 78.06 | 83.23 | 87.10 |
| S-Prompts | 77.63 | 94.65 | 95.98 | 96.37 | 8.50 | 20.89 | 29.21 | 45.10 | 30.54 | 53.96 | 66.90 | 79.74 | 43.23 | 72.26 | **85.81** | 89.68 |
| MaPLe | 75.53 | 94.26 | **96.94** | **98.47** | 10.72 | 28.47 | 38.45 | 55.27 | 37.56 | 67.84 | 77.90 | 85.48 | 38.06 | 74.19 | 84.52 | **92.90** |
| FedTPG | 78.97 | 94.65 | 95.98 | 96.56 | 7.58 | 19.04 | 31.24 | 49.35 | 35.07 | 59.23 | 68.74 | 79.53 | 43.23 | 76.13 | 84.52 | 88.39 |
| Diff-Prompt | **81.45** | 92.35 | 96.18 | 97.71 | **14.42** | **36.97** | **49.35** | **61.92** | **42.91** | **72.38** | **80.13** | **85.74** | **50.97** | **81.94** | 83.23 | 86.45 |

Table 11: Recall Across Different Categories on the Flickr30k val Dataset. (Part 1)

| Method | Other | | | | People | | | | Scene | | | | Vehicles | | | |
|---|---|---|---|---|---|---|---|---|---|---|---|---|---|---|---|---|
| | R@1 | R@5 | R@10 | UB | R@1 | R@5 | R@10 | UB | R@1 | R@5 | R@10 | UB | R@1 | R@5 | R@10 | UB |
| CLIP-Adapter | 32.46 | 57.76 | 65.68 | 73.72 | 68.66 | 92.77 | 94.87 | 96.16 | 29.16 | 65.17 | 77.63 | 84.61 | 66.27 | 84.62 | 89.64 | 91.42 |
| VPT | 30.09 | 57.83 | 65.83 | 73.84 | 62.38 | 89.52 | 93.38 | 95.56 | 35.16 | 68.04 | 76.97 | 83.37 | 51.48 | 81.66 | 89.35 | 92.60 |
| CoOp | 38.00 | 64.86 | 71.99 | 80.94 | 73.80 | **94.87** | 96.46 | 97.73 | 25.11 | 56.95 | 69.67 | 89.24 | 67.75 | **89.64** | **94.38** | **97.04** |
| S-Prompts | 37.45 | 64.16 | 72.26 | 81.39 | 72.65 | **94.87** | **96.87** | **98.31** | 44.29 | 75.41 | 83.82 | 92.24 | 61.83 | 86.39 | 90.53 | 93.20 |
| MaPLe | 41.44 | 67.90 | 75.18 | **83.07** | 72.84 | 94.53 | 96.56 | 97.76 | 51.47 | **81.08** | 88.19 | **94.65** | 61.54 | 86.98 | 92.90 | 96.75 |
| FedTPG | 38.49 | 63.49 | 71.41 | 80.85 | 73.80 | 94.77 | 96.25 | 97.71 | 23.55 | 50.29 | 65.36 | 84.61 | 66.86 | 88.46 | 92.60 | 95.27 |
| Diff-Prompt | **42.57** | **68.27** | **75.67** | 82.80 | **73.90** | 94.86 | 96.51 | 97.87 | **54.01** | 80.95 | **88.52** | 93.74 | **68.05** | 88.46 | 93.79 | 96.15 |

Table 12: Recall Across Different Categories on the Flickr30k val Dataset. (Part 2)

| Method | Animals | | | | Bodyparts | | | | Clothing | | | | Instruments | | | |
|---|---|---|---|---|---|---|---|---|---|---|---|---|---|---|---|---|
| | R@1 | R@5 | R@10 | UB | R@1 | R@5 | R@10 | UB | R@1 | R@5 | R@10 | UB | R@1 | R@5 | R@10 | UB |
| CLIP-Adapter | 73.94 | 91.31 | 92.08 | 94.02 | 4.78 | 14.72 | 20.27 | 34.80 | 32.22 | 55.68 | 64.61 | 76.93 | 50.62 | **80.86** | **85.80** | 88.27 |
| VPT | 63.32 | 86.10 | 90.35 | 93.44 | 5.54 | 19.12 | 28.68 | 42.26 | 26.84 | 52.52 | 62.53 | 76.11 | 33.95 | 57.41 | 67.28 | 70.37 |
| CoOp | 75.68 | 93.05 | 93.82 | 95.37 | 6.12 | 18.16 | 22.75 | 41.49 | 33.52 | 60.75 | 72.33 | 82.87 | 56.17 | 73.46 | 80.25 | 89.51 |
| S-Prompts | 75.48 | 88.99 | 91.51 | 93.24 | 7.27 | 19.89 | 28.68 | 46.27 | 32.91 | 57.24 | 69.60 | 82.09 | 54.32 | 75.31 | 78.40 | 85.80 |
| MaPLe | 76.83 | 89.77 | 92.66 | 94.21 | 8.99 | 30.59 | 41.87 | 55.45 | 39.20 | 71.12 | 80.62 | 86.08 | 48.15 | 74.07 | 77.16 | 87.65 |
| FedTPG | 76.83 | **93.24** | **94.40** | **95.56** | 7.07 | 21.80 | 27.53 | 46.27 | 37.21 | 61.88 | 72.59 | 83.09 | **56.79** | 75.93 | 81.48 | 89.51 |
| Diff-Prompt | **80.50** | 90.35 | 92.28 | 94.02 | **17.21** | **36.14** | **42.83** | **55.64** | **47.35** | **74.28** | **82.18** | **87.60** | 54.32 | 77.16 | 81.48 | **92.59** |

Table 13: Recall Across Different Categories on the Flickr30k test Dataset. (Part 1)

| Method | Other | | | | People | | | | Scene | | | | Vehicles | | | |
|---|---|---|---|---|---|---|---|---|---|---|---|---|---|---|---|---|
| | R@1 | R@5 | R@10 | UB | R@1 | R@5 | R@10 | UB | R@1 | R@5 | R@10 | UB | R@1 | R@5 | R@10 | UB |
| CLIP-Adapter | 33.85 | 60.14 | 67.28 | 76.38 | 71.25 | 92.93 | 95.16 | 96.71 | 28.47 | 63.43 | 73.56 | 81.66 | 77.75 | 91.50 | 94.50 | 95.50 |
| VPT | 29.58 | 55.96 | 64.79 | 73.95 | 64.07 | 90.47 | 94.04 | 96.36 | 32.80 | 63.25 | 73.75 | 80.11 | 66.50 | 84.25 | 87.50 | 90.50 |
| CoOp | 39.45 | 64.34 | 72.97 | 81.51 | 73.80 | 95.42 | 97.26 | 98.36 | 22.24 | 52.99 | 67.76 | 85.61 | 81.00 | 92.50 | 94.25 | 96.75 |
| S-Prompts | 37.82 | 63.99 | 72.32 | 81.59 | 73.16 | 95.40 | 97.56 | 98.53 | 45.95 | 75.54 | 83.01 | 90.86 | 72.00 | 91.50 | 95.00 | 97.00 |
| MaPLe | 40.46 | 66.51 | **75.79** | **83.31** | 74.54 | 95.51 | 97.52 | **98.66** | 49.66 | 79.12 | 86.10 | **92.03** | 73.50 | 91.25 | 93.25 | 96.00 |
| FedTPG | 39.77 | 63.49 | 72.26 | 81.59 | 74.35 | 95.58 | 97.14 | 98.21 | 19.15 | 46.57 | 61.21 | 80.91 | **81.75** | 92.50 | 94.50 | 96.75 |
| Diff-Prompt | **44.61** | **68.02** | 74.39 | 82.34 | **75.21** | **96.06** | **97.68** | 98.44 | **55.65** | **81.04** | **87.03** | 91.66 | 79.75 | **94.25** | **95.75** | **97.25** |

Table 14: Recall Across Different Categories on the Flickr30k test Dataset. (Part 2)

In this part, we conduct a further analysis of the results of different methods on the Flickr30K test and validation datasets. Tab. 11, 12, 13 and 14 presents the R@1, R@5, R@10 and UB scores for different categories, which include Animals, Bodyparts, Clothing, Instruments, Other, People, Scene, and Vehicles. From the figure, we can conclude that Diff-Prompt performs well across all categories, indicating more stable training. It steadily improves performance across different categories without significantly increasing performance in some at the expense of others.

# G CATEGORY-WISE ACCURACY

**Category-wise Accuracy.** In this section, we conduct a further analysis of the results from the quantitative analysis. Specifically, we divide the RefCOCO test dataset into 12 categories based on the "super category" field in the COCO annotations: person, vehicle, outdoor, animal, accessory, sports, kitchen, food, furniture, electronic, appliance, and indoor. We compared the R@1 and R@5 metrics of GLIP-T(A), S-Prompts, MaPLe, FedTPG, and Diff-Prompt across these categories. Metrics for Different Categories for Flickr30k is provided in Appendix F.

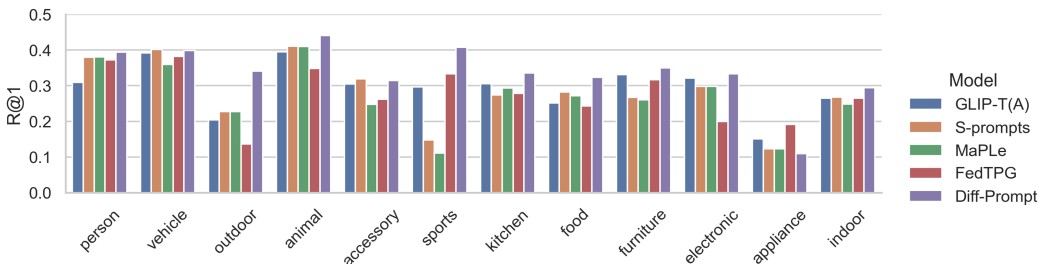

Figure 10: Category-wise R@1 for RefCOCO Val Dataset.

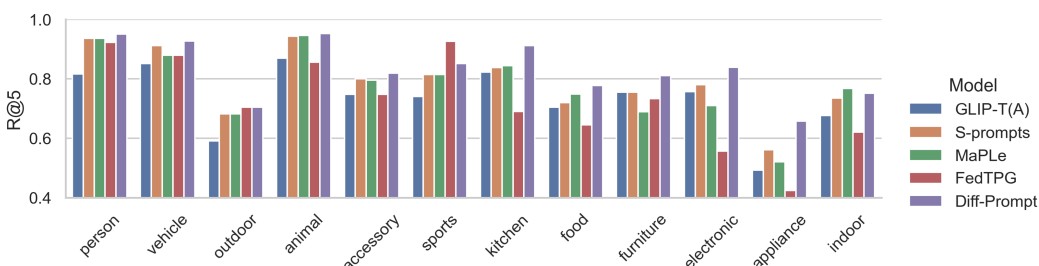

Figure 11: Category-wise R@5 for RefCOCO Val Dataset.

The results are shown in Fig. 10 and 11. From the figures, we can see that, compared to the foundation model, Diff-Prompt shows a more balanced improvement across all categories. This is because the prompt generator, during the second phase of training, can provide a general range for the objects, and the generated prompts do not significantly affect the backbone network. The process of first training the prompt generator and then aligning it helps to effectively prevent overfitting during training. FedTPG generates prompts using a attention layers, but R@1 performance in the outdoor and electronic categories is worse than that of GLIP-T(A). This is because the training results are biased towards certain data, making it unable to achieve improvements across all categories. For the S-Prompts and MaPLe methods, there is a notable performance increase in the person category, while in other categories, their performance shows a slight increase or decrease compared to GLIP-T(A), indicating that they mainly fit the data in the person category.

## H LIMITATION

In this section, we discuss the limitations of our proposed method. First, due to the constraints of the DiT model, our model can only process images with an input size of 224x224, which limits the diversity of the image inputs. A solution is to perform some downsampling and interpolation operations on the image. Second, Diff-Prompt uses a diffusion model to generate prompts in multiple steps, requiring the pretrained VAE and DiT to have strong generalization capabilities; otherwise, performance on specific data may be particularly poor. Additionally, the multi-step generation process of the diffusion model consumes a large amount of time and computational resources. Overall, while Diff-Prompt generates rich, fine-grained prompt information through the diffusion model, it is also influenced by the model itself, leading to high computational complexity.

For the prompt generator, we only concatenate visual and textual information. For future work, we believe more in-depth research on the diffusion model could explore controlled mask generation to produce more fine-grained prompts. As for the design of the prompt generator, exploring other one-step, lightweight generation models to produce prompts could help address the limitations of this paper. Meanwhile, we will explore the capabilities of Diff-Prompt on more downstream tasks, such as PNG (Ding et al., 2022) and GQA (Hudson & Manning, 2019).

