# OpenReview forum: "Diff-Prompt: Diffusion-Driven Prompt Generator with Mask Supervision"
_ICLR.cc/2025/Conference — ICLR 2025 Poster_

### Official Review · Reviewer_A4tj · 2024-11-02

**Soundness:** 3
**Presentation:** 2
**Contribution:** 2
**Rating:** 6
**Confidence:** 4

**Summary:**

This paper introduces the Diffusion-driven Prompt Generator (Diff-Prompt), which leverages the diffusion model to generate detailed and comprehensive prompt information for intricate downstream tasks.

**Strengths:**

This paper is well organized
The performances are competitive

**Weaknesses:**

(1)	The definition in DEEP PROMPTING is confused. What is the definition of E_{i}
(2)	What mechanism do you use to the D latent features of generated prompt? It is not clear
(3)	In subsection 3.1, this study is linked to referring expression comprehension. Nonetheless, there is a lack of information on how the proposed method generates a position to accurately locate the target object described in the expression.
(4)	The advantage of using the generated prompt is not clear. Why can we replace the generated prompts with learnable prompts

**Questions:**

see above-mentioned issues

---

> ### Author Response · Authors · 2024-11-21
> **Response to Reviewer A4tj**
>
> Thank you for your thorough review and for acknowledging the effectiveness of our experimental results. Below, I will provide further clarification regarding the weaknesses you highlighted.
> > The definition in DEEP PROMPTING is confused. What is the definition of $E_{i}$
>
> We apologize for any confusion caused by the lack of clarity in our initial description. Thank you for pointing it out. Deep prompting refers to the use of prompts in the deeper layers of the modality encoder. Take GLIP as an example: both the vision and language encoders consist of 12 transformer layers. The image is first **divided into multiple patches**, which are then **embedded into $E_{0}^{v}$**. The text is **tokenized** and **embedded to produce $E_{0}^{l}$**. These embeddings are then input into the modality encoder. For the i-th transformer layer, the input is $E_{i}$, and the output is $E_{i+1}$. In deep prompting, prompts are concatenated with $E_{i}$ in the first $D$ transformer layers, serving as the new input to each transformer layer. We have added detailed explanations in **Section 3.1** of the updated version of the paper.
>
> > What mechanism do you use to the D latent features of generated prompt? It is not clear
>
> Conditioned on an image with shape [3, 224, 224] and its corresponding caption, the prompt generator randomly samples Gaussian noise with shape [4, 28, 28]. This Gaussian noise is then denoised through multiple steps to obtain **latent prompts** with shape [4, 28, 28], which **correspond to $\widetilde{z}_i$ in the Figure 3**. The latent prompts are in dense form. Considering that the prompt generator and GLIP may operate in different data distributions, we first use the **VAE decoder** in Stage 2 to **reconstruct the latent prompts**, generating prompts (**generated prompts in Figure 3**) of the same size as the image. These generated prompts form a saliency map, which informs the foundation model about which parts of the image require more attention. Subsequently, we design a vision adapter and a language adapter to **map the generated prompts**, i.e., the saliency map, into different **modality encoders**, as shown in Figure 2. In other words, we convert the [224, 224] sized generated prompt into prompts with shapes [4, 96] and [4, 768]. These prompts are then concatenated with $E_i$ and global prompts, and fed into the transformer layer. We have added detailed explanations in **Section 4.2** and **Section 4.3** of the updated version of the paper.
>
> > In subsection 3.1, this study is linked to referring expression comprehension. Nonetheless, there is a lack of information on how the proposed method generates a position to accurately locate the target object described in the expression.
>
> In the GLIP model, visual and textual features are first extracted from the image and text using the vision and language encoders, respectively. The visual features of the image are then input into the **Region Proposal Network (RPN)**. RPN generates multiple anchor boxes by sliding a window over the image feature map, where each anchor box predicts a foreground/background classification and bounding box regression values to adjust its coordinates. Next, RPN applies Non-Maximum Suppression (NMS) to remove redundant and low-confidence proposals, ultimately outputting a set of high-quality candidate regions. We have added detailed explanations in **Section 3.1** of the updated version of the paper.
>
> > The advantage of using the generated prompt is not clear. Why can we replace the generated prompts with learnable prompts
>
> From a broader perspective, generated prompts have significant advantages over directly learnable prompts. Generated prompts can be designed based on prior knowledge, making them more **controllable and interpretable**. For example, segmentation information, edge features, or other task-specific information can be used as prompts, thereby clearly defining the semantic meaning they carry. In this paper, we use masks as prompts and employ a diffusion model to generate task-guided prompt information. In contrast, directly learnable prompts are optimized **end-to-end**, resembling a **"black box" process**, where it is unclear what knowledge has been learned. This lack of interpretability is unacceptable for scenarios where prompts must have explicit meanings. Similar to how large language models require precise prompts to produce desired outputs, prompts themselves should carry **clear and interpretable semantics**.
>
> From a practical application perspective, this approach allows us to design more customized prompts. As indicated by the result in Table 1, our proposed method achieves relatively strong performance.

---

> ### Author Response · Authors · 2024-11-25
>
> Dear Reviewer A4tj,
>
> Thank you once again for dedicating your time and effort to reviewing our work. We would greatly appreciate it if you could let us know whether our response has sufficiently addressed your concerns. With the rebuttal phase nearing its conclusion, we look forward to your feedback and are available to provide any further clarification you might need.

---

> > ### Comment · Area_Chair_QNxW · 2024-12-03
> >
> > Dear Reviewer A4tj,
> >
> >       As the deadline of discussion period is today, please respond to the author reply to see if your concerns are well-addressed at your earliest convenience. Thank you very much.
> >
> > Best,
> > AC

---

### Official Review · Reviewer_LEnk · 2024-11-02

**Soundness:** 2
**Presentation:** 3
**Contribution:** 2
**Rating:** 8
**Confidence:** 4

**Summary:**

This paper introduces Diff-Prompt, a novel prompt learning approach that leverages diffusion models to generate rich prompts for multimodal tasks. Diff-Prompt employs a three-stage process: first training a Mask-VAE to compress masks into latent space, then using an improved Diffusion Transformer to generate prompts in this latent space with mask supervision, and finally aligning the denoising process with the pre-trained model in semantic space. The method addresses limitations of existing prompt learning approaches by generating more informative and task-specific prompts.

**Strengths:**

I find the approach to be innovative and a departure from other methods in prompt learning for improved generalization. The paper is well presented, and easy to follow. The method achieves strong performance on two vision-language understanding datasets, RefCOCO and Flickr30k compared to existing methods.

**Weaknesses:**

I have two major concerns regarding this work.

1. The authors do not report on other generalization benchmarks such as the ones reported by prompt learning techniques like CoOP and Maple, which the authors compare with. These are representative tasks in generalization to novel classes, new target datasets and unseen domain shifts. Why were these benchmarks not considered, since methods reporting on these benchmarks are considered in the evaluation?

2. The authors are missing comparisons with multiple recent works related to prompt learning for generalization such as [1,2,3,4,5,6]. Why were these methods not considered during evaluation and comparison?

[1] Z. Li et al., "PromptKD: Unsupervised Prompt Distillation for Vision-Language Models", in CVPR, 2024.

[2] M. U. Khattak et al., "Self-regulating Prompts: Foundational Model Adaptation without Forgetting," ICCV, 2023.

[3] J. Hasan et al., "Align Your Prompts: Test-Time Prompting with Distribution Alignment for Zero-Shot Generalization," NeurIPS, 2023.

[4] S. Roy and A. Etemad, ‘Consistency-guided Prompt Learning for Vision-Language Models’, in ICLR, 2024.

[5] Y. Zhang, C. Zhang, K. Yu, Y. Tang, and Z. He, ‘Concept-Guided Prompt Learning for Generalization in Vision-Language Models’, in AAAI, 2024.

[6] Y. Zhang, K. Yu, S. Wu, and Z. He, ‘Conceptual Codebook Learning for Vision-Language Models’, in ECCV, 2024.

**Questions:**

Please refer to the weaknesses. In general I am concerned with evaluation settings which are different compared to other major works in prompt learning, as well as missing comparisons to recent works.

---

> ### Author Response · Authors · 2024-11-21
> **Response to Reviewer LEnk Part 1**
>
> Thank you for recognizing the innovation of our method and acknowledging the effectiveness of our experiments. Regarding the weaknesses you mentioned, I will provide a detailed response below.
>
> >The authors do not report on other generalization benchmarks such as the ones reported by prompt learning techniques like CoOP and Maple, which the authors compare with. These are representative tasks in generalization to novel classes, new target datasets and unseen domain shifts. Why were these benchmarks not considered, since methods reporting on these benchmarks are considered in the evaluation?
>
> Thank you for pointing out the lack of generalization benchmarks in our experiments. In our method, we focused on addressing complex downstream tasks but overlooked comparisons on more commonly used benchmarks. To address this, we have supplemented experiments on **cross-dataset generalization** and **cross-domain generalization**, and included comparisons with the methods highlighted by the reviewer. The results have been added to the updated version in **Section 5.4 Ablation Study under Cross-Domain Generalization and Cross-Dataset Generalization**. The specific results are shown in the table below.  The experimental results show that, while our method does not achieve state-of-the-art performance, it still demonstrates considerable generalization ability.
>
> Comparison with SOTA Methods on Cross-Domain Evaluation
> | Source      | ImageNet | -V2   | -S    | -A    | -R    | Avg.  |
> |-------------|----------|-------|-------|-------|-------|-------|
> | CLIP        | 66.73    | 60.83 | 46.15 | 47.77 | 73.96 | 57.18 |
> | CoOp        | 71.51    | 64.20 | 47.99 | 49.71 | 75.21 | 59.28 |
> | MaPLe       | 70.72    | 64.07 | 49.15 | 50.90 | 76.98 | 60.27 |
> | PromptKD    | -        | -     | -     | -     | -     | -     |
> | PromptSRC   | 71.27    | 64.35 | 49.55 | 50.90 | 77.80 | 60.65 |
> | CoPrompt    | 70.80    | 64.25 | 49.43 | 50.50 | 77.51 | 60.42 |
> | CPL         | 73.53    | 65.18 | 49.92 | 50.73 | 77.38 | 60.80 |
> | CoCoLe      | **73.88**| **65.86** | 50.89 | **51.75** | **78.89** | **61.85** |
> | Diff-Prompt | 72.06    | 64.29 | **51.06** | 50.97 | 77.18 | 60.88 |
>
> Comparison with SOTA Methods on Cross-Dataset Evaluation Part 1
> | Source       | ImageNet | Caltech101 | OxfordPets | StanfordCars | Flowers102 | Food101 |
> |--------------|----------|------------|------------|--------------|------------|---------|
> | CoOp         | 71.51    | 93.70      | 89.14      | 64.51        | 68.71      | 85.30       |
> | MaPLe        | 70.72    | 93.53      | 90.49      | 65.57        | 72.23      | 86.20      |
> | PromptKD     | -        | 93.61      | **91.59**  | **73.93**    | 75.33      | **88.84**|
> | PromptSRC    | 71.27    | 93.60      | 90.25      | 65.70        | 70.25      | 86.15    |
> | CoPrompt     | 70.80    | 94.50      | 90.73      | 65.67        | 72.30      | 86.43     |
> | CPL          | 73.53    | 95.52      | 91.64      | 66.17        | 73.35      | 87.68         |
> | CoCoLe       | **73.88**| **95.88**  | 91.93      | 67.79        | 74.17      | 87.97   |
> | Diff-Prompt  | 72.06    | 94.63      | 91.08      | 66.85        | **75.57**  | 87.26     |
>
> Comparison with SOTA Methods on Cross-Dataset Evaluation Part 1
> | Source        | Aircraft | SUN397 | DTD   | EuroSAT | UCF101 | Average |
> |--------------|----------|------------|------------|--------------|------------|---------|
> | CoOp           | 18.47    | 64.15  | 41.92  | 46.39   | 66.55  | 63.88   |
> | MaPLe         | 24.74    | 67.01  | 46.49  | 48.06   | 68.69  | 66.30   |
> | PromptKD    | 26.24    | 68.57  | **55.08**| **63.74**   | **76.39**| **71.33** |
> | PromptSRC  | 23.90    | 67.10  | 46.87  | 45.50   | 68.75  | 65.81   |
> | CoPrompt     | 24.00    | 67.57  | 47.07  | 51.90   | 69.73  | 67.00   |
> | CPL              | 27.36    | 68.24  | 48.96  | 51.25   | 70.52  | 68.07   |
> | CoCoLe        | 28.83    | 68.75  | 49.26  | 52.75   | 72.78  | 68.91   |
> | Diff-Prompt   | **29.07**| **69.03**| 48.87 | 52.83| 73.32  | 68.85   |
>
> It is worth noting that PromptAlign [3] represents a test-time adaptation approach, which operates in a different application scenario compared to the other methods. Therefore, we did not include it in direct comparisons but discussed it in the related work section.

---

> ### Author Response · Authors · 2024-11-21
> **Response to Reviewer LEnk Part 2**
>
> > The authors are missing comparisons with multiple recent works related to prompt learning for generalization such as [1,2,3,4,5,6]. Why were these methods not considered during evaluation and comparison?
>
> Thank you for providing the suggested methods. In the generalization ablation experiments, we have supplemented comparisons with these methods. Regarding the comparison on the RES task, we found that these methods cannot be effectively transferred, and the reasons are as follows:
>
> The GLIP model uses Swin as the vision encoder and BERT as the language encoder. Its weights have been fine-tuned for downstream tasks, which makes the GLIP encoder's output incompatible with computing logits in the same way as CLIP. Key components in these methods, such as the distillation loss in PromptKD, $L_{\text{SCL-logits}}$ in PromptSRC, $L_{\text{entropy}}$ in PromptAlign, $L_{\text{ce}}$ in CoPrompt, and the key-value caching strategies in CPL and CoCoLe, cannot be seamlessly applied to the GLIP model. For GLIP or other complex downstream task models, designing different prompt strategies is more appropriate.
>
> Considering the lack of comparisons with the latest methods, we have supplemented our experiments with a recently designed prompt tuning method **VFPT** (NeurIPS 2024)[1] and several adapter tuning methods, **Tip-Adapter** (ECCV 2022)[2], **Meta-adapter** (NeurIPS 2023)[3], and **MMA** (CVPR 2024)[4]. The results demonstrate that our proposed method still shows certain performance advantages. We have added these experiments and conclusions to **Table 1 and Section 5.2** in the updated version.
>
> | Method       | RefCOCO (testA) |       |       | RefCOCO (testB) |       |       | RefCOCO (val) |       |       |
> | ------------ | --------------- | ----- | ----- | --------------- | ----- | ----- | ------------- | ----- | ----- |
> |              | R@1             | R@5   | UB    | R@1             | R@5   | UB    | R@1           | R@5   | UB    |
> | Tip-adapter  | 34.68           | 86.24 | 97.25 | 32.83           | 78.39 | 94.02 | 34.56         | 83.06 | 95.47 |
> | Meta-adapter | 35.02           | 87.96 | 98.64 | 33.29           | 78.67 | 95.14 | 36.54         | 85.09 | 96.34 |
> | MMA          | 36.68           | 89.03 | 99.13 | 34.67           | 79.06 | 95.88 | 35.28         | 86.46 | 97.18 |
> |VFPT         |37.24 |91.45 |98.23 |31.98| 79.36 |97.74| 34.92| 87.93| 98.41|
> | Diff-Prompt  | 39.08           | 94.71 | 99.63 | 36.09           | 85.67 | 99.00 | 37.94         | 90.55 | 99.37 |
>
> | Method       | Flickr30k (test) |       |       | Flickr30k (val) |       |       |
> | ------------ | ---------------- | ----- | ----- | --------------- | ----- | ----- |
> |              | R@1              | R@5   | UB    | R@1             | R@5   | UB    |
> | Tip-adapter  | 50.16            | 74.89 | 84.53 | 48.22           | 73.54 | 85.19 |
> | Meta-adapter | 51.32            | 75.36 | 85.16 | 49.18           | 74.87 | 88.14 |
> | MMA          | 52.60            | 77.04 | 85.77 | 51.43           | 76.28 | 89.46 |
> |VFPT         | 55.82          |  76.16|  88.67| 51.53| 75.94| 88.31|
> | Diff-Prompt  | 59.53            | 81.85 | 90.46 | 57.39           | 81.20 | 90.54 |
>
> [1] Zeng R, Han C, Wang Q, et al. Visual Fourier Prompt Tuning[J]. arXiv preprint arXiv:2411.01327, 2024.
>
> [2] Zhang R, Zhang W, Fang R, et al. Tip-adapter: Training-free adaption of clip for few-shot classification[C]//European conference on computer vision. Cham: Springer Nature Switzerland, 2022: 493-510.
>
> [3] Song L, Xue R, Wang H, et al. Meta-adapter: An online few-shot learner for vision-language model[J]. Advances in Neural Information Processing Systems, 2023, 36: 55361-55374.
>
> [4] Yang L, Zhang R Y, Wang Y, et al. MMA: Multi-Modal Adapter for Vision-Language Models[C]//Proceedings of the IEEE/CVF Conference on Computer Vision and Pattern Recognition. 2024: 23826-23837.

---

> ### Author Response · Authors · 2024-11-25
>
> Dear Reviewer LEnk,
>
> Thank you once again for dedicating your time and effort to reviewing our work. We would greatly appreciate it if you could let us know whether our response has sufficiently addressed your concerns. With the rebuttal phase nearing its conclusion, we look forward to your feedback and are available to provide any further clarification you might need.

---

> > ### Comment · Reviewer_LEnk · 2024-11-26
> >
> > Dear Authors,
> >
> > First of all, I would like to thank you for providing such detailed comparisons in such a short time. Right now I would say I am not completely convinced. The response part 1 shows that your method does not generalize that well compared to other prompt learning methods. However, part 2 of the response does show that your method performs well in certain settings in comparison to specific methods.
> >
> > Therefore, I will wait to hear from other reviewers before making a final decision.

---

> ### Author Response · Authors · 2024-11-26
> **Further Explanation on Generalization Ability**
>
> Thank you for your response. Regarding your concerns about the generalization ability of our method, I would like to provide further clarification.
> - First, our method is **less suited for image classification tasks**. This is because Diff-Prompt relies on **paired images and captions to generate prompts**. However, in image classification tasks, each class name is encoded, and a prompt is generated for every image paired with each class name, resulting in a **quadratic increase** in computational cost. For example, in ImageNet, which has 1000 classes, a single image would need to generate prompts with 1000 textual features. To address this in our experiments, we averaged the 1000 textual features, which inevitably **diluted the representation of the text**.
>
> - Second, our method demonstrates good generalization ability, though it has not yet reached the state-of-the-art (SOTA) level. This is because our method is **not specifically designed for generalization**; instead, it focuses on solving **complex, multimodal, fine-grained downstream tasks**.
>
> Currently, methods with strong generalization capabilities, such as CoPrompt and CoCoLe, achieve this by designing **regularization terms**. CPL and CoCoLe, for example, enhance generalization through the **use of caches**. Therefore, achieving strong generalization often requires specific tricks. In contrast, our method is centered around **prompt design**, with the primary purpose of guiding the model to focus on specific parts of the image based on textual input. This approach provides significant advantages for visual understanding tasks.
>
> Our prompts can thus be considered specifically designed for complex visual understanding tasks, which cover a wide range of applications. From the results, Diff-Prompt demonstrates superior accuracy across various datasets compared to other prompt-based methods.
>
> - Furthermore, Diff-Prompt can be **seamlessly integrated with regularization or cache-based techniques to achieve stronger generalization capabilities**. Notably, these regularization terms and cache-based methods typically operate on the outputs of the encoder, whereas Diff-Prompt provides guidance during the encoding process itself. This ensures that our method does not affect the encoder’s outputs, maintaining their integrity while enhancing the performance.
>
>
> - Finally, these methods are evaluated using a 16-shot setting and trained for a small number of epochs (set to 5 in our experiments) on ImageNet. This indicates that the improvement in accuracy from generalization experiments is limited by the model's fitting ability on the ImageNet dataset. Complex network structures typically require more extensive training. To further investigate the model's generalization capability, it is more reasonable to evaluate it using zero-shot methods. As shown in **Appendix D**, we explored the zero-shot capability of models trained on the RefCoCo dataset and conducted zero-shot experiments on 10 unseen object detection tasks. While the GLIP model itself has strong generalization capabilities, the experimental results demonstrate that Diff-Prompt enhances the generalization ability of GLIP-T(A) to some extent.

---

> > ### Comment · Area_Chair_QNxW · 2024-12-03
> >
> > Dear Reviewer LEnk,
> >
> >       As the deadline of discussion period is today, please respond to the author reply to see if your further concerns are well-addressed at your earliest convenience. Thank you very much.
> >
> > Best,
> > AC

---

### Official Review · Reviewer_3hUM · 2024-11-03

**Soundness:** 3
**Presentation:** 3
**Contribution:** 3
**Rating:** 6
**Confidence:** 3

**Summary:**

The paper introduces Diff-Prompt, a diffusion-driven prompt generator designed to enhance fine-tuning of pre-trained multimodal models. Traditional prompt learning methods often fail on complex tasks due to limited prompt richness and specificity. Diff-Prompt addresses this by using a three-stage process: (1) compressing mask information into latent space with a Mask-VAE, (2) employing an enhanced Diffusion Transformer (DiT) to generate prompts in the latent space with mask supervision, and (3) aligning these generated prompts with a pre-trained model's semantic space for fine-tuning. Experimental results show that Diff-Prompt significantly outperforms other methods on pixel-level tasks, achieving notable improvements in recall metrics (R@1 and R@5).

**Strengths:**

1. The use of a Mask-VAE and diffusion-driven prompt generator enables the creation of rich prompts, which improves the model's ability to effectively capture information in the latent space.
2. The alignment of generated prompts with the pre-trained model in the semantic space ensures better guidance for the pre-trained model, enhancing performance during fine-tuning.
3. The experimental results on a complex fine-grained multimodal downstream task validate the effectiveness of the proposed method, showcasing its applicability to challenging scenarios.

**Weaknesses:**

Compared to other methods, Diff-Prompt has slightly more learnable parameters (5.5M). More parameters may require more memory and lead to longer inference time.

**Questions:**

1. The vision and language adaptors adopt the same architecture. How to reasonably exploit the advantages of vision and language cues to ensure consistent fine-tuning of the pre-trained model?
2. What potential strategies could be employed to address the high computational complexity and image input size limitations of the Diff-Prompt model?

---

> ### Author Response · Authors · 2024-11-21
> **Response to Reviewer 3hUM**
>
> We sincerely appreciate your positive feedback and recognition of the effectiveness of our method and experiments. We would like to address the concerns you have raised as follows:
>
> > The vision and language adaptors adopt the same architecture. How to reasonably exploit the advantages of vision and language cues to ensure consistent fine-tuning of the pre-trained model?
>
> The adapters adopt the same architecture, which first performs a downsampling operation to extract dense prompt information and then upsamples it to map into the corresponding semantic space of each modality. The outputs of the adapters for different modalities are passed to their respective modality encoders, where they **interact with the modality-specific embeddings through attention mechanisms**. As a result, the knowledge learned by the adapters for different modalities is distinct. For example, the vision encoder encodes information from images, and the prompts generated by the vision adapter tend to focus on **spatial information** in the images. In contrast, the language encoder encodes textual information, and the prompts generated by the language adapter are more inclined to **capture semantic or contextual cues**.
>
> > What potential strategies could be employed to address the high computational complexity and image input size limitations of the Diff-Prompt model?
>
> Diff-Prompt has high computational complexity. I believe this issue can be addressed from the following perspectives:
> - **Optimizing the Diffusion Model Design**: The optimization methods for the diffusion model can align with mainstream strategies. This includes employing **distillation techniques** to compress the model size, designing **more efficient sampling strategies**, and exploring other **diffusion acceleration** methods. For example, [1] generates results in a single step using a distillation strategy, from which we can draw inspiration to reduce computational complexity.
> - **Enhancing the Prompt Generator**: From the perspective of the prompt generator, future work could explore **alternative generative models**. In Appendix E (Ablation Study for Prompt Precision), we concluded that the **generated prompts do not need to be exceptionally precise** to provide effective guidance. This suggests that we could trade off some performance for higher efficiency. For instance, we could utilize more mature models such as **VAE or GAN**. Additionally, **traditional machine learning** methods could be considered, such as using the Canny operator to extract image edges as base prompts or leveraging other techniques to extract textual and visual information. However, traditional machine learning methods have limitations, such as difficulty in achieving cross-modal interaction, which requires further research.
>
> The input image is used as a condition for the network and Diff-Prompt generates latent prompts that are decoded back to the original image size using the Mask-VAE decoder. The simplest solution is to constrain the model's input and output sizes and resize the output results as needed. However, this approach limits the model's generalization capability. A more optimal solution is as follows:
>
> (1) **Improve Mask-VAE Compatibility Across Resolutions**: Our Mask-VAE encoder and decoder are composed of convolutional layers and activation functions. Therefore, as long as the input image's width and height are multiples of 8 (a limitation of the model due to the need for upsampling and downsampling), the Mask-VAE can encode and decode the image effectively. To enhance its performance across varying resolutions, the Mask-VAE needs to be **trained on images of multiple resolutions**. This ensures robust performance across different resolutions, which serves as the foundation of Diff-Prompt.
>
> (2) **Improving Prompt Generation**: Several enhancements can be considered for the prompt generator in Diff-Prompt: **replacing the linear layers in the DiT model with convolutional layers**, similar to Mask-VAE, to enable dynamic input and output lengths; designing an image condition and output alignment module to further constrain the output based on the input conditions; or dynamically adjusting the size of the initial noise based on the input image, allowing the model to better adapt to varying input resolutions.
>
> [1] Luo W, Huang Z, Geng Z, et al. One-step diffusion distillation through score implicit matching[J]. arXiv preprint arXiv:2410.16794, 2024.

---

> ### Author Response · Authors · 2024-11-25
>
> Dear Reviewer 3hUM,
>
> Thank you once again for dedicating your time and effort to reviewing our work. We would greatly appreciate it if you could let us know whether our response has sufficiently addressed your concerns. With the rebuttal phase nearing its conclusion, we look forward to your feedback and are available to provide any further clarification you might need.

---

> > ### Comment · Area_Chair_QNxW · 2024-12-03
> >
> > Dear Reviewer 3hUM,
> >
> >       As the deadline of discussion period is today, please respond to the author reply to see if your concerns are well-addressed at your earliest convenience. Thank you very much.
> >
> > Best,
> > AC

---

### Official Review · Reviewer_cYSP · 2024-11-03

**Soundness:** 3
**Presentation:** 2
**Contribution:** 2
**Rating:** 6
**Confidence:** 5

**Summary:**

The paper introduces Diff-Prompt, a diffusion-based prompt generation method aimed at enhancing fine-tuning for complex multimodal tasks. Unlike conventional prompt tuning methods, which struggle with fine-grained tasks due to limited prompt richness, Diff-Prompt uses a multi-stage process with a Mask-VAE and an improved Diffusion Transformer to generate detailed, task-specific prompts. This approach outperforms other fine-tuning methods, achieving notable improvements in recall and accuracy on tasks requiring nuanced multimodal understanding.

**Strengths:**

-The motivation of this paper is very clear and interesting, making it an engaging topic worth exploring in depth.

-The illustrations in the paper are very clear, and the validation on the visual grounding task is quite comprehensive.

-There is a significant improvement in some metrics for the referring expression comprehension task.

-The code has been released, which I believe will be very helpful for further study.

**Weaknesses:**

1. Generalizability needs further verification: Why was testing only done on GLIP? How would it perform with other models? Why was the evaluation limited to the referring expression comprehension task? What about tasks like RES and PNG for segmentation? Would it also be effective for other more complex tasks? If it only works with this model and task, then the contribution is quite limited, and the paper's title should be adjusted to specify that it is focused on the REC task.

2. The introduction of diffusion slows inference speed by twofold, which is a significant cost. Could this be optimized?

3. In Tables 1 and 2, the improvement in the UB metric compared to MaPLe is minimal, despite MaPLe being a much simpler method. Why?

4. The effectiveness of using diffusion in this paper warrants deeper analysis. Since the core contribution of the method itself isn’t very substantial, the primary value lies in using diffusion to generate prompts. Exploring the underlying mechanism further could provide more profound insights.

**Questions:**

1. Generalizability needs further verification: Why was testing only done on GLIP? How would it perform with other models? Why was the evaluation limited to the referring expression comprehension task? What about tasks like RES and PNG [1] for segmentation? Would it also be effective for other more complex tasks? If it only works with this model and task, then the contribution is quite limited, and the paper's title should be adjusted to specify that it is focused on the REC task.

[1] PPMN: Pixel-Phrase Matching Network for One-Stage Panoptic Narrative Grounding.

---

> ### Author Response · Authors · 2024-11-21
> **Response to Reviewer cYSP Part 1**
>
> Thank you for your insightful feedback on our paper. We sincerely appreciate your recognition of the motivation behind our approach, as well as your acknowledgment of our experimental results and the value of our work for further exploration. We are eager to address your concerns and questions and look forward to providing clarifications where needed.
>
> >Generalizability needs further verification: Why was testing only done on GLIP? How would it perform with other models? Why was the evaluation limited to the referring expression comprehension task? What about tasks like RES and PNG [1] for segmentation? Would it also be effective for other more complex tasks? If it only works with this model and task, then the contribution is quite limited, and the paper's title should be adjusted to specify that it is focused on the REC task.
>
> Thank you for pointing out the limitations of our method in terms of generalization, particularly noting that it was only evaluated on the GLIP model for the REC downstream task. In response, we have conducted additional experiments to address this concern.
>
> First, we sincerely apologize for our lack of familiarity with the PNG task, which made it challenging to identify a suitable backbone and adapt our method within a short timeframe. As an alternative, we selected two other downstream tasks—**Referring Expression Segmentation (RES)** and **Visual Question Answering (VQA)**—to further validate our approach. Regarding the PNG task, we have included it in our future work to explore the generalization capabilities of our method more comprehensively. As part of future work, we first need to train a foundation model for the PNG task. This model uses an encoder to extract features from both images and text, which are then passed to a downstream head to generate results for the PNG task. Additionally, the encoder needs to adopt a transformer architecture to allow for the dynamic addition of any number of prompts. This has also been explicitly acknowledged in the limitations section of the revised version of our paper.
>
> For the RES task, we use the CLIPSeg model, and for the VQA task, we employ the BLIP model. Detailed explanations of these experiments are provided in the "Generalization Across Downstream Tasks and Backbones" section of the ablation study in the updated version. The results, presented in the table below, show that our method improves performance for both models. Notably, the selected models are already extensively trained on their respective datasets (e.g., CLIPSeg was thoroughly trained on the PhaseCut dataset). Despite this, our method is still able to achieve further accuracy improvements. This effectiveness stems from our method's ability to guide the model to **focus on relevant parts of the image based on captions**, making it highly applicable to **various visual understanding tasks**.
>
> This part has been added to **Section 5.4 Generalization Across Downstream Tasks and Backbones** in the updated version.
>
> Result for Referring Expression Segmentation
>
> | Method                           | mIoU | IoU_FG | AP   |
> | -------------------------------- | ---- | ------ | ---- |
> | CLIPSeg(PC)                      | 46.1 | 56.2   | 78.2 |
> | CLIPSeg(PC, D=128)               | 48.2 | 56.5   | 78.2 |
> | CLIPSeg(PC) + Diff-Prompt        | 47.8 | 56.4   | 78.2 |
> | CLIPSeg(PC, D=128) + Diff-Prompt | 49.6 | 57.0   | 78.7 |
>
> Result for Visual Question Answering
>
> | Method                                  | VQA (test-dev) | VQA (test-std) | NLVR² (dev) | NLVR² (test-P) |
> | --------------------------------------- | -------------- | -------------- | ----------- | -------------- |
> | BLIP                                    | 78.24          | 78.17          | 82.48       | 83.08          |
> | BLIP$_\text{CapFilt-L}$               | 78.25          | 78.32          | 82.15       | 82.24          |
> | BLIP + Diff-Prompt                      | 78.59          | 78.88          | 82.94       | 83.76          |
> | BLIP$_\text{CapFilt-L}$ + Diff-Prompt | 78.92          | 79.24          | 83.09       | 84.06          |
>
> To more comprehensively evaluate the generalization capability of our method, we also conduct experiments on two more commonly used benchmarks: **cross-domain generalization** and **cross-dataset generalization**, both of which involve **image classification tasks**. These experiments are detailed in the **Cross-Domain Generalization** and **Cross-Dataset Generalization** sections of the **Section 5.4** ablation study in the updated version. The results demonstrate that our method maintains strong generalization capabilities across these benchmarks as well.

---

> ### Author Response · Authors · 2024-11-21
> **Response to Reviewer cYSP Part 2**
>
> > The introduction of diffusion slows inference speed by twofold, which is a significant cost. Could this be optimized?
>
> The optimization can be considered from the following aspects:
> - **Optimizing the Diffusion Model**: The optimization direction for the diffusion model aligns with the general advancements in diffusion models. This includes improving **sampling strategies** and **distillation strategies**. Recent studies[1] have demonstrated that diffusion models can generate results in a single step. By leveraging such approaches, we could significantly accelerate the inference speed of the diffusion model.
> - **Alternative Prompt Generators**: From the perspective of the prompt generator, we could explore alternative models that generate results in a single step, such as **GANs or VAEs**. While the generated quality may be slightly inferior, it would still suffice for prompt generation. Additionally, **traditional machine learning methods** could be employed to extract high-frequency or edge information from images, which can serve as the base prompts.
>
> > In Tables 1 and 2, the improvement in the UB metric compared to MaPLe is minimal, despite MaPLe being a much simpler method. Why?
>
> The UB metric reflects the **potential of the model** to identify the correct target across all predictions, and as such, the room for further improvement in UB is limited for both MaPLe and Diff-Prompt. This is because UB largely represents the **upper limit of the foundation model's capabilities**. Prompt learning methods are designed to maximize the foundation model's inherent performance but **cannot surpass its intrinsic upper limit**. Compared to MaPLe, Diff-Prompt achieves an improvement of nearly 3% in both R@1 and R@5, which demonstrates the superiority of our approach.
>
> >The effectiveness of using diffusion in this paper warrants deeper analysis. Since the core contribution of the method itself isn’t very substantial, the primary value lies in using diffusion to generate prompts. Exploring the underlying mechanism further could provide more profound insights.
>
> Firstly, most existing works design **automatically learned prompts**, leaving it unclear what the model ultimately learns. We believe that our primary contribution is the design of a visual prompt generation method, which differs from traditional automated learning strategies by being more **controllable and meaningful**—a direction that has been largely unexplored in previous works.
>
> Now, let us explain why we chose to use diffusion models for prompt generation:
>
> 1. **Rich Information Representation**: Extensive cutting-edge research has demonstrated that diffusion models can generate rich and detailed information. This means that, for models of the same size, diffusion models can generate more information. Moreover, while producing the same amount of information, diffusion models maintain a minimal model size, with the trade-off being computational complexity. Given our focus on semantic richness, we opts for diffusion models.
> 2. **Key Benefits of Diffusion-Generated Prompts**: Diffusion models play a critical role in multimodal information fusion. For the vision encoder, the generated prompts indicate which visual embeddings to focus on. For the language encoder, the generated prompts capture semantic and contextual information, effectively guiding the model's attention.
> 3. **Alignment of Diffusion and Encoder Processes**: In our experiments, we align the denoising process of the diffusion model with the encoding process of the encoder. The results demonstrate the effectiveness of this approach. The underlying principle is that the diffusion denoising process can be viewed as a **progressive information fusion process**, while the encoder encodes sparse data into dense representations. This hierarchical alignment enables the encoder to better perceive the modality interaction process, thereby enhancing overall model performance.
>
> [1] Luo W, Huang Z, Geng Z, et al. One-step diffusion distillation through score implicit matching[J]. arXiv preprint arXiv:2410.16794, 2024.

---

> ### Author Response · Authors · 2024-11-25
>
> Dear Reviewer cYSP,
>
> Thank you once again for dedicating your time and effort to reviewing our work. We would greatly appreciate it if you could let us know whether our response has sufficiently addressed your concerns. With the rebuttal phase nearing its conclusion, we look forward to your feedback and are available to provide any further clarification you might need.

---

> > ### Comment · Area_Chair_QNxW · 2024-12-03
> >
> > Dear Reviewer cYSP,
> >
> > As the deadline of discussion period is today, please respond to the author reply to see if your concerns are well-addressed at your earliest convenience. Thank you very much.
> > Best, AC

---

### Official Review · Reviewer_PtHg · 2024-11-08

**Soundness:** 2
**Presentation:** 1
**Contribution:** 1
**Rating:** 3
**Confidence:** 5

**Summary:**

To address the limitation of prompt learning in applying to complex tasks requiring high granularity, the authors propose a three-stage training prompt generator named Diff-Prompt. This method aims to generate rich prompts and provide sufficient information for pretrained models in fine-grained downstream tasks. Through experiments on two vision-language understanding datasets and comparisons with other methods across visual aspects and various metrics, they demonstrate the superiority of their approach.

**Strengths:**

The paper's comparative experiments include multiple models and provide statistical results, offering guidance for other researchers interested in prompt learning.
The proposed Diff-Prompt surpasses other existing efficient fine-tuning methods on multiple evaluation metrics, which may stimulate further thinking among researchers.
The authors train the prompt generator through mask supervision, controlling the generation of fine edge information via mask variations, which is a novel idea.

**Weaknesses:**

Diff-Prompt is compared with two different types of efficient parameter tuning methods, but the number of baselines for the two methods is seriously imbalanced. It would be preferable to include more comparative experiments with adapter methods.
Compared with previous work, this paper seems focused on training masks through diffusion models to generate masks with fine edges for boundary control, but the contribution beyond this aspect is insufficient.
The authors claim to generate rich prompts, but the masks generated during the diffusion process do not exhibit finely controllable generation. This approach may not necessarily adapt to highly fine-grained tasks, which is evidently inconsistent with the problem the authors intend to solve.

**Questions:**

The authors mentioned that Diff-Prompt may suffer from poor generalization. Based on this, have they proposed any improvement plans to mitigate this issue and enable reproduction by other researchers?
How does the strategy of Diff-Prompt differ from independently training a conditional mask generator and then performing information interaction with a multimodal encoder?
During prompt generation, did the authors consider how to maintain a balance between visual and textual information to avoid information imbalance caused by modality bias?

---

> ### Author Response · Authors · 2024-11-21
> **Response to Reviewer PtHg Part 1**
>
> Thank you for your detailed review and for recognizing the experimental contributions and innovative aspects of our work. We have carefully reviewed the weaknesses and questions you raised and have provided responses to each of them below.
>
> >Diff-Prompt is compared with two different types of efficient parameter tuning methods, but the number of baselines for the two methods is seriously imbalanced. It would be preferable to include more comparative experiments with adapter methods.
>
> Thank you for pointing this out. Indeed, we initially lacked comparisons with Adapter Tuning methods. To address this, we have added **three adapter tuning methods**: **Tip-Adapter** (ECCV 2022)[1], **Meta-Adapter** (NeurIPS 2023)[2], and **MMA** (CVPR 2024)[3]. Tip-Adapter employs a training-free adapter tuning approach, which adjusts data distribution solely by updating key-value pairs without the need for training. Meta-Adapter introduces meta-learning into adapter tuning, while MMA designs a multimodal adapter that fully considers interactions between modalities. Tip-adapter and Meta-adapter are methods frequently used for comparison in adapter tuning, while MMA is a very recent and highly effective approach.
>
> The results are shown in the table below. We have also updated the original manuscript and merged the two previous comparison tables into a single table, which can be found as **Table 1** in the revised version.
>
> | Method       | RefCOCO (testA) |       |       | RefCOCO (testB) |       |       | RefCOCO (val) |       |       |
> | ------------ | --------------- | ----- | ----- | --------------- | ----- | ----- | ------------- | ----- | ----- |
> |              | R@1             | R@5   | UB    | R@1             | R@5   | UB    | R@1           | R@5   | UB    |
> | Tip-adapter  | 34.68           | 86.24 | 97.25 | 32.83           | 78.39 | 94.02 | 34.56         | 83.06 | 95.47 |
> | Meta-adapter | 35.02           | 87.96 | 98.64 | 33.29           | 78.67 | 95.14 | 36.54         | 85.09 | 96.34 |
> | MMA          | 36.68           | 89.03 | 99.13 | 34.67           | 79.06 | 95.88 | 35.28         | 86.46 | 97.18 |
> | Diff-Prompt  | 39.08           | 94.71 | 99.63 | 36.09           | 85.67 | 99.00 | 37.94         | 90.55 | 99.37 |
>
> | Method       | Flickr30k (test) |       |       | Flickr30k (val) |       |       |
> | ------------ | ---------------- | ----- | ----- | --------------- | ----- | ----- |
> |              | R@1              | R@5   | UB    | R@1             | R@5   | UB    |
> | Tip-adapter  | 50.16            | 74.89 | 84.53 | 48.22           | 73.54 | 85.19 |
> | Meta-adapter | 51.32            | 75.36 | 85.16 | 49.18           | 74.87 | 88.14 |
> | MMA          | 52.60            | 77.04 | 85.77 | 51.43           | 76.28 | 89.46 |
> | Diff-Prompt  | 59.53            | 81.85 | 90.46 | 57.39           | 81.20 | 90.54 |
>
> From the results, it can be observed that prompt tuning methods generally achieve greater improvements in accuracy compared to adapter tuning methods. This is because adapter tuning typically requires adjusting the learned network to fit the **entire data distribution**, which may **compromise the generalization ability of the original backbone network**, thereby making training more challenging. As a result, the accuracy improvement is relatively limited.
>
> [1] Zhang R, Zhang W, Fang R, et al. Tip-adapter: Training-free adaption of clip for few-shot classification[C]//European conference on computer vision. Cham: Springer Nature Switzerland, 2022: 493-510.
>
> [2] Song L, Xue R, Wang H, et al. Meta-adapter: An online few-shot learner for vision-language model[J]. Advances in Neural Information Processing Systems, 2023, 36: 55361-55374.
>
> [3] Yang L, Zhang R Y, Wang Y, et al. MMA: Multi-Modal Adapter for Vision-Language Models[C]//Proceedings of the IEEE/CVF Conference on Computer Vision and Pattern Recognition. 2024: 23826-23837.

---

> ### Author Response · Authors · 2024-11-21
> **Response to Reviewer PtHg Part 2**
>
> > Compared with previous work, this paper seems focused on training masks through diffusion models to generate masks with fine edges for boundary control, but the contribution beyond this aspect is insufficient.
>
> Thank you for your feedback. I would like to further elaborate on the contributions of our method:
>
> (1) **Generalization to various visual unstanding tasks**: Diff-Prompt can be applied to a wide range of text-based visual understanding tasks, such as referring expression, referring expression segmentation, visual question answering, image captioning, visual reasoning, and more. The concept behind our method can also be extended to other modalities, such as visual-audio classification, as the prompts we design are based on adding attention from other modalities to the visual modality, offering a wide range of application scenarios.
>
> (2)  **Better Interpretability**: Prompt learning initially emerged in the natural language processing domain, where natural language prompts (e.g., CoT) were used to guide models. While randomly learned prompts can improve performance to some extent, what the model actually learns remains unclear. This lack of interpretability hinders deeper research, and in practical applications, interpretable prompts are preferable. Similarly, in the computer vision, most existing visual prompt learning methods rely on end-to-end random optimization, which is a **black-box process**. Our goal is to generate specific prompts, using these as the **foundation** to guide pre-trained models. Therefore, our contribution is not to generate masks but to use masks as a **prompt foundation** to make the approach more interpretable.
>
> (3) **Multimodal Alignment**: Current interaction methods rely on network designs to enable communication between modalities. This is simple but effective. In our work, we conduct another exploration, that is to align the **denoising process** in diffusion models with the **encoding process** of the pre-trained model. Experimental results show that this alignment is effective and leads to improved performance.
>
> (3) **Complex Downstream Tasks**: Unlike prior prompt learning research, which mainly focuses on simpler tasks like image classification, our work applies prompt learning to more complex multimodal tasks, such as referring expression comprehension, which requires precise object localization. We also analyze multiple metrics to provide a more comprehensive evaluation of our method’s performance.
>
> (4) **Boundary Control**: Our work does not aim to generate fine edges for boundary control but rather to produce a **saliency map**. The distinction is that boundary control requires high precision, while a saliency map simply informs the model about **which regions of the image require more attention**. Since the saliency map functions as a prompt rather than a precise segmentation result, pixel-level accuracy is unnecessary. Our ablation studies show that even a coarse saliency map can effectively guide the model, demonstrating the robustness of our approach. Additionally, when we refer to "fine-grained" tasks, we mean downstream tasks like referring expression comprehension, where the final output demands precise four-pixel bounding box coordinates.
>
> (5) **Outstanding performance**: Our experiments demonstrate that generating prompts using customized methods also delivers excellent performance compared to randomly learned prompt learning methods.

---

> ### Author Response · Authors · 2024-11-21
> **Response to Reviewer PtHg Part 3**
>
> >The authors claim to generate rich prompts, but the masks generated during the diffusion process do not exhibit finely controllable generation.
>
> I will explain the controllability of our method from the following two perspectives:
> - **Controllability of the Diffusion Model Generation Process**. In the design of the prompt generator, the image and text are concatenated after passing through the embedding layer and used as a condition to guide the generation of the diffusion model, enabling control over the generated results. This approach also ensures that the generated prompts are highly correlated with both modalities.
> - **Average Recall for IoU Intervals**. For the RefCOCO dataset, where each caption corresponds to a bounding box and a segmentation mask, we conducted the following experiment: we performed an in-depth analysis of the results on the RefCOCO validation dataset by reconstructing the prompts generated by the prompt generator. We then calculated the Intersection over Union (IoU) between the reconstructed masks and the ground-truth masks in the labels. Based on the IoU values, we divided all results into 10 bins, such as IoU > 0 and ≤ 0.1, IoU > 0.1 and ≤ 0.2, and so on, and calculated the accuracy for each bin. The results are shown in the table below. We conduct a more detailed analysis in **Appendix E Ablation Study for Prompt Precision** of the updated version.
>
> | IoU  | 0-0.1  | 0.1-0.2 | 0.2-0.3 | 0.3-0.4 | 0.4-0.5 | 0.5-0.6 | 0.6-0.7 | 0.7-0.8 | 0.8-0.9 | 0.9-1.0 | AVG    |
> | ---- | ------ | ------- | ------- | ------- | ------- | ------- | ------- | ------- | ------- | ------- | ------ |
> | R@1  | 0.242 | 0.296  | 0.328  | 0.358  | 0.386  | 0.394  | 0.402  | 0.413  | 0.415  | 0.419  | 0.379 |
>
>
> From the results, we can observe that when IoU is below 0.4, the accuracy significantly improves as IoU increases. However, when IoU exceeds 0.4, the improvement rate slows down, and when IoU exceeds 0.8, the accuracy stabilizes. This indicates that **even a coarse prompt, without requiring a highly precise mask, can provide effective guidance**. This finding aligns with the essence of prompts, which is to serve as guidance rather than the final result.
>
> > This approach may not necessarily adapt to highly fine-grained tasks, which is evidently inconsistent with the problem the authors intend to solve.
>
> The above statements, along with the results in **Table 1**, demonstrate that the prompts proposed by our method do not need to be highly precise to effectively guide fine-grained downstream tasks. To further illustrate this point, we added two additional downstream tasks: Referring Expression Segmentation and Visual Question Answering, as shown in **Table 4** and **Table 5** of the revised version. The experimental results confirm the effectiveness of our method.
>
> Result for Referring Expression Segmentation
>
> | Method                           | mIoU | IoU_FG | AP   |
> | -------------------------------- | ---- | ------ | ---- |
> | CLIPSeg(PC)                      | 46.1 | 56.2   | 78.2 |
> | CLIPSeg(PC, D=128)               | 48.2 | 56.5   | 78.2 |
> | CLIPSeg(PC) + Diff-Prompt        | 47.8 | 56.4   | 78.2 |
> | CLIPSeg(PC, D=128) + Diff-Prompt | 49.6 | 57.0   | 78.7 |
>
> Result for Visual Question Answering
>
> | Method                                  | VQA (test-dev) | VQA (test-std) | NLVR² (dev) | NLVR² (test-P) |
> | --------------------------------------- | -------------- | -------------- | ----------- | -------------- |
> | BLIP                                    | 78.24          | 78.17          | 82.48       | 83.08          |
> | BLIP$_\text{CapFilt-L}$               | 78.25          | 78.32          | 82.15       | 82.24          |
> | BLIP + Diff-Prompt                      | 78.59          | 78.88          | 82.94       | 83.76          |
> | BLIP$_\text{CapFilt-L}$ + Diff-Prompt | 78.92          | 79.24          | 83.09       | 84.06          |

---

> ### Author Response · Authors · 2024-11-21
> **Response to Reviewer PtHg Part 4**
>
> > The authors mentioned that Diff-Prompt may suffer from poor generalization. Based on this, have they proposed any improvement plans to mitigate this issue and enable reproduction by other researchers?
>
> Our method's generalization capability is not poor. In the original Appendix D, we stated: **"However, compared to GLIP-L, which has a larger parameter size and more training data, Diff-Prompt and GLIP-L still have a significant gap."** However, **Diff-Prompt use GLIP-T(A) as foundation model**. Our findings indicate that the zero-shot performance of our method is **not inferior to GLIP-T(A)**, though there is a **noticeable gap compared to GLIP-L**. Throughout the paper, we have consistently used **GLIP-T(A) as the foundation model**, so a fair comparison would be with GLIP-T(A). However, we also included a comparison with GLIP-L to serve as an upper bound for the zero-shot capability of the model. The significant performance gap between the original GLIP-T(A) and GLIP-L can be attributed to several factors. Compared to GLIP-T(A), GLIP-L has a much larger model size (GLIP-L’s model weights are 6.9GB, whereas GLIP-T(A)’s are only 2.43GB) and is trained on more extensive datasets, which endows it with stronger generalization capabilities.
>
> Furthermore, to more comprehensively evaluate the generalization ability of our proposed method, we conducted comparisons using more general benchmarks, as presented in the updated version's **Section 5.4 ablation study** on **cross-domain generalization** and **cross-dataset generalization**.
>
> Notably, methods like PromptSRC, CoPrompt, CPL, and CoCoLe are **specifically designed for generalization tasks**, utilizing regularization terms or caches to achieve generalization capability. In contrast, methods such as CoOp, MaPLe, and Diff-Prompt focus on the design of prompts. Our proposed method demonstrates competitive performance compared to state-of-the-art methods, showcasing that our approach also possesses a certain degree of generalization capability. Parts of the results are shown in the table below. For detailed results, please refer to the updated version.
>
> | Source      | ImageNet | -V2   | -S    | -A    | -R    | Avg.  |
> |-------------|----------|-------|-------|-------|-------|-------|
> | CLIP        | 66.73    | 60.83 | 46.15 | 47.77 | 73.96 | 57.18 |
> | CoOp        | 71.51    | 64.20 | 47.99 | 49.71 | 75.21 | 59.28 |
> | MaPLe       | 70.72    | 64.07 | 49.15 | 50.90 | 76.98 | 60.27 |
> | PromptKD    | -        | -     | -     | -     | -     | -     |
> | PromptSRC   | 71.27    | 64.35 | 49.55 | 50.90 | 77.80 | 60.65 |
> | CoPrompt    | 70.80    | 64.25 | 49.43 | 50.50 | 77.51 | 60.42 |
> | CPL         | 73.53    | 65.18 | 49.92 | 50.73 | 77.38 | 60.80 |
> | CoCoLe      | 73.88    | 65.86 | 50.89 | 51.75 | 78.89 | 61.85 |
> | Diff-Prompt | 72.06    | 64.29 | 51.06 | 50.97 | 77.18 | 60.88 |
>
> To better improve the generalization ability of the model, **a general-purpose prompt generator** is essential, as the generated prompts determine the foundation for fine-tuning. In future work, we could consider training a universal prompt generator or using traditional machine learning methods to extract features from images to generate prompts.
>
> As for reproducibility, we have already provided an anonymous repository for the code. We will release the model weights and continue to update the code in the future.
>
> > How does the strategy of Diff-Prompt differ from independently training a conditional mask generator and then performing information interaction with a multimodal encoder?
>
> - **Rich Expressive Capability**: Our model leverages the rich expressive capability of diffusion models, which enables it to generate fine-grained and detailed representations. Furthermore, by incorporating multimodal information during the design process, it ensures a comprehensive and effective approach, making it both innovative and forward-looking.
> - **Unique Modality Alignment**: We align the denoising process of the diffusion model with the encoding process of the encoder, effectively simulating a multimodal information fusion process. This alignment provides a distinctive and efficient approach to integrating multimodal data.
> - **Lightweight and Efficient**: While ensuring the quality of the generated prompts, we have minimized the model's parameters. The final model has a FP32 storage size of only 309MB, and in practical applications, using FP16 computation can further reduce memory overhead. On the contrary, the smallest ViT-B SAM model is 358MB and cannot incorporate language modality information.
>
> In summary, we have developed a customized lightweight prompt generator that not only effectively considers multimodal information but also generates rich prompts while occupying minimal memory, making it efficient and practical.

---

> ### Author Response · Authors · 2024-11-21
> **Response to Reviewer PtHg Part 5**
>
> > During prompt generation, did the authors consider how to maintain a balance between visual and textual information to avoid information imbalance caused by modality bias?
>
> The prompts generated by the prompt generator incorporate information from both modalities, focusing on the location of the object described in the text within the image. The absence or insufficiency of information from either modality would make it impossible to approximate the object's position. A prompt is meaningful only if it can effectively indicate the object's approximate location; otherwise, it may mislead the pre-trained model and result in decreased accuracy. Therefore, the closer the reconstructed prompt is to the mask, the better it reflects the **degree of modality fusion**, indicating **sufficient integration of information from both modalities**. The effectiveness of our method further validates that our generated prompts are indeed useful, demonstrating the **successful fusion of information from different modalities**.

---

> ### Author Response · Authors · 2024-11-25
>
> Dear Reviewer PtHg,
>
> Thank you once again for dedicating your time and effort to reviewing our work. We would greatly appreciate it if you could let us know whether our response has sufficiently addressed your concerns. With the rebuttal phase nearing its conclusion, we look forward to your feedback and are available to provide any further clarification you might need.

---

> > ### Comment · Area_Chair_QNxW · 2024-12-03
> >
> > Dear Reviewer PtHg,
> >
> > As the deadline of discussion period is today, please respond to the author reply to see if your concerns are well-addressed at your earliest convenience. Thank you very much.
> >
> > Best,
> > AC

---

### Author Response · Authors · 2024-11-21
**Summary of Changes in the Rebuttal Revision**

We appreciate the reviewers' time and valuable feedback. In response to their comments and suggestions, we have made several revisions to the paper. All changes have been highlighted in blue in the updated version. A summary of the updates is provided below:
- In Sections 3.1 and 3.2, we provide further explanations of the preliminaries of the method.
- In Sections 4.2 and 4.3, we provide a more detailed description of the method design and a more in-depth explanation of the principles behind these design choices.
- In Sections 5.1 and 5.2, we **supplement several state-of-the-art efficient fine-tuning methods**, including three adapter tuning methods: Tip-adapter, Meta-adapter, and MMA. Additionally, we introduce one prompt tuning method, VFPT, and analyzed the corresponding experimental results.
- In Section 5.4, we conduct an in-depth exploration of the model's **generalization capabilities** through three experiments: **Cross-dataset Generalization**, **Cross-domain Generalization**, and **Generalization Across Downstream Tasks and Backbones**. The first two experiments are commonly used generalization benchmarks, while the third experiment examines the performance of Diff-Prompt across different backbones and downstream tasks.
- In Appendix E, we add an experiment analyzing the variation of **Recall across different IoU intervals**.
- We move the category-wise accuracy from the in-depth analysis to the Appendix H.

---

### Author Response · Authors · 2024-11-21
**For All Reviewers**

We sincerely thank all the reviewers for their insightful comments and unanimous recognition of the contributions of our work:
- **Novel and interesting motivation**: We appreciate the Reviewer  **PtHg**, **cYSP**, **LEnk** recognition of the novelty of our work and their acknowledgment that it is an intriguing and worthwhile direction for further exploration.
- **Effective method and superior performance**: We express our gratitude to **ALL** for their positive feedback, acknowledging the effectiveness of our method and the strong experimental results.

We will now address the key arguments raised in the reviews and provide detailed responses to each reviewer. We kindly encourage the reviewers to refer to the **updated version** of our paper.

Once again, we sincerely thank the reviewers for their time and effort in evaluating our work. We look forward to any additional insights or questions that may arise from our research.

---

### Meta-Review · Area_Chair_QNxW · 2024-12-26

**Metareview:**

In this work, the authors present Diff-Prompt, a novel three-stage prompt generation method based on diffusion models, designed to enhance the fine-tuning of pretrained multimodal models by providing richer and more fine-grained information. Extensive experiments on referring expression comprehension demonstrate that the proposed method outperforms other parameter-efficient fine-tuning approaches, highlighting its effectiveness. The majority of reviewers agree that the paper is well-written, and the proposed method is novel and interesting (PtHg, cYSP, LEnk). They acknowledge the strong experimental results and the effectiveness of the proposed approach. However, some reviewers raise concerns, requesting additional experiments and analyses with more baselines and tasks to further validate the method’s effectiveness and generalization capabilities (PtHg, cYSP, LEnk). Other concerns include the slow inference speed of the diffusion model (cYSP, 3hUM) and the need for clearer explanations of the mathematical notations and contributions (A4tj). The authors provided detailed responses, addressing most of the reviewers’ concerns and significantly revising the manuscript. As a result, reviewers LEnk, cYSP, and A4tj increased their ratings from negative to positive, while reviewers PtHg and 3hUM did not participate in the final discussion. The paper ultimately received an average score of 5.8, with one accept, three borderline accepts, and one reject. I also agree that the proposed method has its own merits and recommend the paper for acceptance. Moreover,  the authors should follow the suggestions from reviewers LEnk, cYSP, and A4tj by incorporating all materials discussed during the rebuttal and including additional comparisons with learnable prompts in the final version.

**Additional Comments On Reviewer Discussion:**

During the rebuttal, the authors provided detailed responses, including additional experiments on various tasks (e.g., visual question answering and referring expression segmentation) beyond referring expression comprehension, demonstrating the method’s generalization capability. They also conducted comparisons with more baselines (e.g., Tip-Adapter (ECCV 2022), Meta-Adapter (NeurIPS 2023), and MMA (CVPR 2024)) and offered clearer explanations of the proposed method. As a result, reviewers LEnk (i.e., rating is increased from 5 to 8), cYSP (5 to 6), and A4tj (5 to 6) change their evaluation scores from negative to positive. Reviewers PtHg and 3hUM, however, did not participate in the final discussion. Most reviewers gave the paper positive ratings, except reviewer PtHg. Given that PtHg did not engage in the discussion, I recommend to accept the paper.

---

### Decision · Program_Chairs · 2025-01-22

Accept (Poster)